# Proteomic signatures of smoking and their associations with risk of incident diseases and mortality in diverse populations

Sihao Xiao[1,2,3] ✉, Bowen Liu[1], M. Austin Argentieri [4,5], Lazaros Belbasis[1], Claire L. Shovlin [6,7], Jennifer A. Collister [1], Siyi Wang[8], Eilis Hannon [8], Jun Liu [1], Kahung Chan [1], Rami Muath Mosaoa[3,9], Liming Li[10,11,12], Jun Lv [10,11,12], Canqin Yu[10,11,12], Dianjianyi Sun [10,11,12], Jonathan Mill [8], Robert Clarke [1], David J. Hunter[1,5], Derrick Bennett[1], Alejo J. Nevado-Holgado[2,3,13], Zhengming Chen [1], Najaf Amin [1] & Cornelia M. van Duijn [1,2,3] ✉

Smoking is the most important behavioural determinant of morbidity and mortality. Using machine learning on plasma levels of 2,917 proteins in the UK Biobank (n = 43,914), we develop a proteomic Smoking Index (pSIN) comprising 51 proteins that accurately distinguish current from never smokers (AUC = 0.95; 95% CI 0.94–0.95). Validation in the China Kadoorie Biobank (n = 3,977) shows similar accuracy (AUC = 0.91; 95% CI 0.89–0.92). pSIN is significantly associated with the risk of all-cause mortality and 18 major chronic diseases, including cardiovascular, renal, pulmonary, neurodegenerative, and cancer outcomes. Among current and former smokers, pSIN predicts death and 11 diseases independently of self-reported smoking history and lifestyle factors. Genome-wide analysis identifies 125 genes (e.g., *ALPP*, *CST5*, *IL12B*) associated with pSIN, while exposome analysis highlights maternal smoking, diet, physical activity, and air pollution as key modifiers. Notably, pSIN tracks recovery among former smokers and identifies those whose disease risks remain comparable to current smokers. These findings demonstrate that plasma proteomics effectively capture the biological imprint of smoking and predict smoking-related morbidity and mortality, offering a more nuanced, molecularly grounded assessment of individual variation in biological response to smoking.

Seventy years after the British Doctors Study first demonstrated an increased risk of death from lung cancer[1], myocardial infarction and chronic obstructive pulmonary disease (COPD)[2] among smokers, smoking still accounts for about 14% of total deaths and 8% of disability-adjusted life years globally[3]. Indeed, smoking is associated with higher risks of numerous diseases across different organ systems, including most cancers, respiratory diseases, cardiovascular diseases, and diseases of the liver, brain, kidney, and bladder, among others[4]. While smoking is often initiated in youth, smoking-related diseases typically manifest in middle age or later. Throughout the life course, smoking behaviour, including the type and quantity of tobacco consumed, may fluctuate, and many individuals attempt to quit smoking

multiple times without success due to its addictive nature. Additionally, as smoking is becoming socially unacceptable in an increasing number of societies, the validity of smoking history is further compromised[5,6]. Consequently, there is an urgent need to develop objective measures that assess smoking history and possible differences in smoking behaviour dynamics, including the recovery status of previous smokers. Such biomarkers will provide more accurate measurements of smoking exposure and will help to identify molecular mechanisms linking smoking with disease risks.

Exhaled carbon monoxide (CO) and plasma cotinine levels are widely used to validate smoking status[7,8], but both capture smoking status within the last 24 h, with limited ability to characterise long-term smoking habits and health effects and to reveal mechanisms of action associated with smoking. There have been major advances in our understanding of the epigenomic signatures of cigarette smoking during the last decade, particularly using smoking-related DNA methylation[9]. A meta-analysis of 16 epigenome-wide studies demonstrated smoking-induced DNA methylation of 2623 CpGs annotated to 1405 genes among current smokers[10]. Of these, 185 CpGs showed persistent alterations in previous smokers, and 36 CpGs showed persistent alterations for up to 30 years after smoking cessation. While DNA methylation markers reflect smoking history in smokers and previous smokers, DNA methylation-based smoking profiles have only been associated with the risks of a limited number of conditions, including COPD, lung cancer, stroke and all-cause mortality, independent of smoking history[11–13].

Circulating proteins, however, may reflect not only smoking exposure as captured by DNA methylation but also the cumulative biological effects of responses to cigarette smoking (including oxidative stress or other mechanisms) or the biological responses to limit the hazards of smoking[9,14,15]. Thus, plasma proteomic profiles may enhance the ability to quantify the direct effects and the physiological responses to smoking and ultimately to predict the risk of subsequent morbidity and mortality among current and previous smokers.

In this work, to address these limitations in the field, we develop a proteomic-based smoking profile in the UK Biobank (UKB) ($n = 43,914$) using machine learning with high accuracy Area Under the Curve (AUC = 0.95; CI:0.94–0.95). We further externally validate the accuracy of our proteomic-based smoking score in participants from the China Kadoorie Biobank (CKB) ($n = 3977$, AUC = 0.91; CI:0.89–0.92). We then explore the genetic and exposome factors influencing the smoking proteomic score. Finally, we study the associations between our proteomic-based smoking score and clinical risk factors, blood-based biomarkers, risks of 27 major diseases and mortality in the UKB. These findings demonstrate that plasma proteomics can be used to effectively capture smoking patterns and risk of smoking-related morbidity and mortality, and that proteomics-based estimation of smoking history provides more nuanced information about differences between individuals in the dynamics of biological response to smoking.

## Results
To model the relationships between the plasma proteome and cigarette smoking, we used a subset of the UKB with available measurements of the plasma proteome. The study population consisted of 43,914 participants, with 4732 self-reported as current regular cigarette smokers, and 23,778 as never smokers. Baseline characteristics of study participants in the UKB are provided in Supplementary Data 1. We observed significant age ($p = 3.44 \times 10^{-26}$) and sex ($p = 1.64 \times 10^{-71}$) differences between current and never smokers. Therefore, we regressed out age and sex from the expression of each protein to eliminate their effects for downstream analyses. For external validation, we used a subset of the CKB, in which the plasma proteome was measured using an identical OLINK assay panel (Supplementary Data 1 for baseline characteristics).

## Proteomic signatures of smoking
In a training dataset comprising 70% of UKB participants, we developed a gradient boosting tree model using plasma protein levels as features to discriminate current smokers from never smokers (Fig. 1). The model achieved a high area under receiver operating characteristic (ROC) curve (AUC) after 5-fold cross-validation (CV) within the training dataset (mean AUC = 0.96, SD = 0.004; F1 = 0.98, SD = 0.006; Averaged precision (AP) = 0.93, SD = 0.006; Balanced accuracy (BA) = 0.90, SD = 0.006) (Supplementary Fig. 1a) and showed similar performance in the 30% UKB holdout test dataset (AUC = 0.95, SD = 0.004, F1 = 0.98, SD = 0.004; AP = 0.91, SD = 0.006; BA = 0.88, SD = 0.006). After using the Boruta algorithm to identify all proteins that contribute to predicting smoking status, we found that 51 of 2917 proteins discriminated current smokers and never smokers (98.3% reduction in protein size) with a mean AUC of 0.95 in 5-fold CV in the training dataset (SD = 0.005) (F1 = 0.97, SD = 0.005; AP = 0.93, SD = 0.004, BA = 0.90, SD = 0.004) (Supplementary Fig. 1b and Supplementary Data 3) and an AUC of 0.95 (SD = 0.004) in the test dataset (F1 = 0.97, SD = $1.11 \times 10^{-16}$; AP = 0.91, SD = $3.33 \times 10^{-16}$; BA = 0.89, SD = $1.11 \times 10^{-16}$) (Fig. 2a). Due to the imbalance of the participant numbers between current smokers and never smokers (1:5), we also calculated the scoring matrix after undersampling the test dataset. When case to control ratio is 1:1, the AUC stays the same at 0.95 (SD = $2.22 \times 10^{-16}$) (F1 = 0.99, SD = $1.11 \times 10^{-16}$; AP = 0.97, SD = $2.22 \times 10^{-16}$; BA = 0.89, SD = $3.33 \times 10^{-16}$). This model yielded a sensitivity of 84.6% and a specificity of 95%. We also compared the performance of the model in males and females in the UKB test dataset. The result showed slightly higher performance in females compared to males, with an AUC of 0.96 (SD = $1.11 \times 10^{-16}$) and 0.94 (SD = $2.06 \times 10^{-296}$), respectively. To assess the added value of pSIN compared to using a single biomarker, we evaluated the predictive performance of the top three proteins ranked by SHAP value within the UKB test dataset (Supplementary Fig. 1c). ALPP and CXCL17 demonstrated relatively strong discrimination with AUCs of 0.88 and 0.87, respectively, while ACVRL1 showed lower performance with an AUC of 0.76. However, none of these individual proteins matched the predictive power of pSIN, which achieved an AUC of 0.95. These results highlight the advantage of using a multi-protein composite score over a single biomarker for more accurate stratification of current smokers and never smokers. The UKB model derived from the training set was subsequently externally validated in CKB, achieving an AUC of 0.91 (SD = $2.22 \times 10^{-16}$; Fig. 2a) (F1 = 0.93, SD = $1.11 \times 10^{-16}$; AP = 0.87, SD = $1.11 \times 10^{-16}$; BA = 0.79, SD = $2.22 \times 10^{-16}$), and a sensitivity of 70.8% and specificity of 95%. After undersampling the majority class, the balanced testing dataset achieved a similar AUC of 0.91 (SD = $4.44 \times 10^{-16}$) (F1 = 0.97, SD = $4.44 \times 10^{-16}$; AP = 0.93, SD = $1.11 \times 10^{-16}$; BA = 0.80, SD = $1.11 \times 10^{-16}$). We further compared the performance of the trained model between males and females in CKB. The model showed a slightly higher AUC for males (AUC = 0.90, SD = $2.06 \times 10^{-296}$; F1 = 0.98, SD = $4.44 \times 10^{-16}$; AP = 0.91, SD = $2.06 \times 10^{-296}$; AP = 0.77, SD = $2.22 \times 10^{-16}$) comparing to females (AUC = 0.89, SD = $2.06 \times 10^{-296}$; F1 = 0.98, SD = $2.22 \times 10^{-16}$; AP = 0.90, SD = $1.11 \times 10^{-16}$; AP = 0.74, SD = $2.22 \times 10^{-16}$). We next calculated a predicted score for the overall UKB population, including previous smokers defined as a Proteomic Smoking INdex (pSIN) (Fig. 2c). Although the curves largely overlapped, the mean pSIN was significantly higher in previous smokers (mean = 2.8; SD = 1.8) than in never smokers (mean = −3.7; SD = 1.4, $p < 2.2 \times 10^{-308}$).

Overall, the proteins selected were predominantly involved in biological processes including epithelial cell proliferation (IL12B, IGFBP4, TNFSF12, ACVRL1, MMP12, PRL, KIT), regulation of immune system (IL12B, MMP12, SCGB1A1, MERTK, PDCD1, IL7R), cell growth (IGFBP4, TNR, ACVRL1, SCGB3A1, ISLR2), and T cell activation (IL12B, CD1C, SCGB1A1, KIT, IL7R). Tissue enrichment analysis was performed

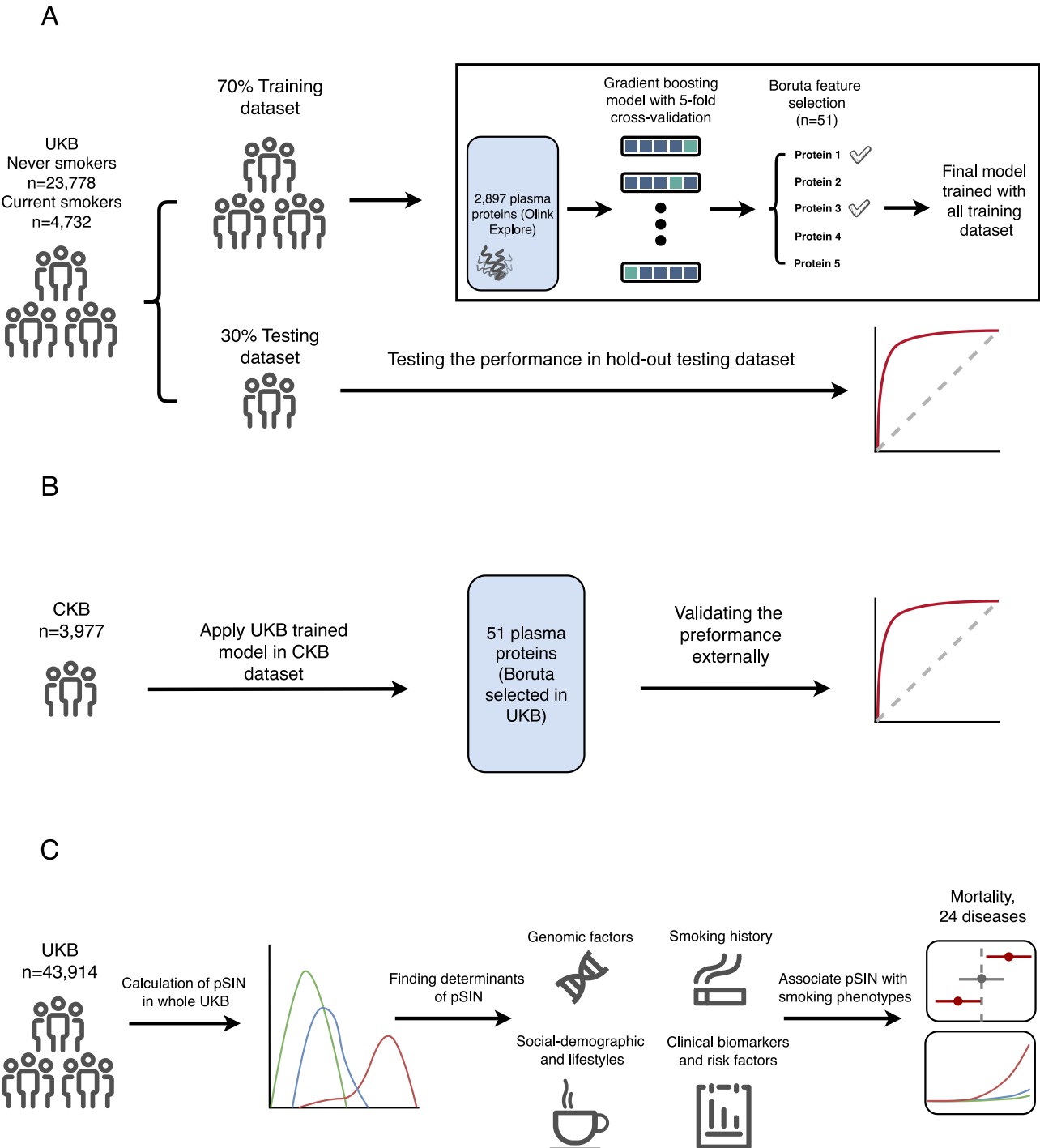

**Fig. 1 | Overview of the study design and analytic approach. A** gradient boosting classification model was built in the 70% randomly selected UKB population, differentiating current smokers and never smokers. Boruta feature selection algorithm was then used to select only relevant features for downstream analysis. **B** The model trained in UKB training dataset was further validated externally in CKB male population. **C** proteomic Smoking INdex (pSIN) was calculated for the whole UKB cohort. Smoking behavioural, genomic, and exposome determinants of pSIN were studied, followed by associating pSIN with incident health outcomes.

among selected proteins using RNA expression data from the Genotype-Tissue Expression (GTEx) project[16]. Results indicated that many of these proteins were differentially expressed in tissues either directly exposed to or affected by smoking, including lung, salivary glands, colon, oesophagus and adipose tissues (Supplementary Fig. 2a, b). The top proteins contributing to the model prediction (Fig. 2b) included ALPP, CXCL17 and ACVRL1, all of which showed higher levels in smokers. All three are predominantly expressed in the lungs, oesophagus and minor salivary glands. ALPP is synthesised in the liver by a

metalloenzyme that catalyses the hydrolysis of phosphoric acid monoesters and was previously found to be associated with COPD and cancers[17]. CXCL17 is a mucosal chemokine that attracts immature dendritic cells and blood monocytes to the lungs and was previously associated with lung cancer[18]. ACVRL1 is an activin receptor-like kinase that has recently been shown to separately mediate transcytosis of low-density lipoprotein into arterial endothelium[19]. Individuals with loss-of-function *ACVRL1* DNA variants have lower rates of progression of atherosclerotic vascular diseases[20].

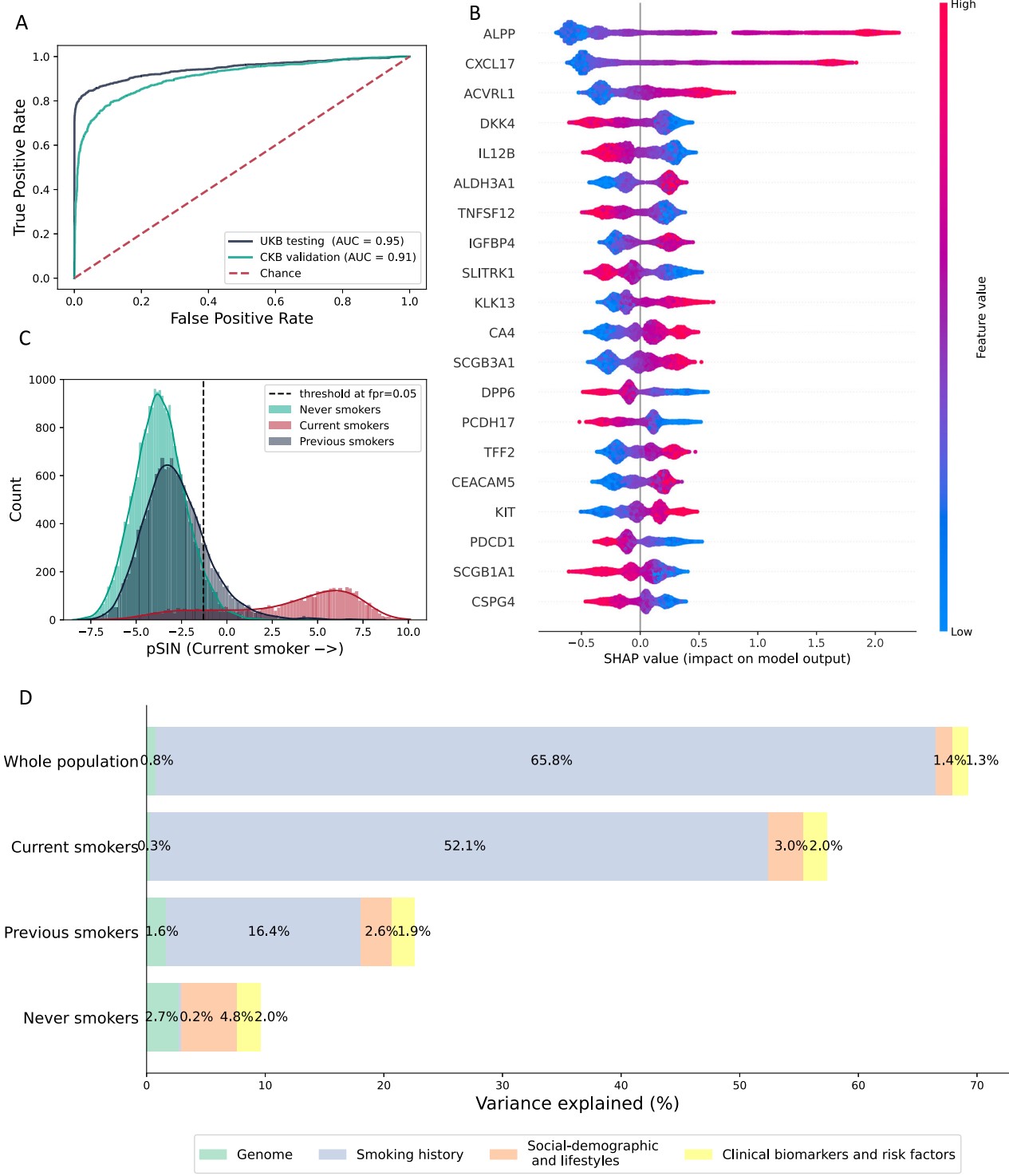

**Fig. 2 | Protein profile differentiating current and never smokers and its determinants. A** shows the performance of the model in UKB 30% left-out testing dataset and CKB external validation dataset. **B** shows a summary plot for the SHAP value of the top 20 selected proteins. Each dot denotes a participant, the colour of the dots denotes the protein expression level, and the X-axis denotes its contribution to the model decision. Proteins were ranked by means of the absolute SHAP value. **C** shows the distribution of pSIN in the whole UKB population. Dotted line denotes the cut-off value at FPR 0.05 when differentiating current and never smokers. **D** Four gradient boosting models were employed to assess the contribution of intrinsic and extrinsic factors to pSIN. The additional variance explained by each category was calculated by subtracting the variance explained by the previous model from the variance explained by the current model.

## Determinants and correlates of pSIN

pSIN is predominantly determined by smoking history, as the number of years smoked, cigarettes smoked per day, and pack years that were smoked were each positively correlated with pSIN scores in both current and previous smokers (Supplementary Data 4). Among

previous smokers, a longer duration since cessation was strongly associated with lower pSIN scores. Analysis of the data in 5-year intervals indicated a continuous decline in pSIN scores up to and exceeding 30 years of smoking cessation, suggesting ongoing recovery from regular smoking-related effects (Supplementary Data 4). In

individuals who had never smoked, passive exposure to smoking was correlated with higher pSIN scores, demonstrating its high sensitivity. Similar trends were observed in CKB current smokers, where smoking duration, smoking exposure, and smoking intensity were all significantly and positively associated with pSIN. However, among former smokers, significant associations were observed only for years since smoking cessation and smoking duration, whereas associations with smoking intensity and smoking exposure were not statistically significant. This may be because there are more modifiers related to the difference in behavioural or epidemiological characteristics between the Chinese and UK populations, such as relapse after quitting smoking in previous smokers than in current smokers. (Supplementary Data 5).

To understand the genetic architecture underlying the biological consequences of smoking pathology, as indexed by pSIN, we performed a genome-wide association study (GWAS) of pSIN in the UKB. We applied a linear mixed model in SAIGE, adjusting for age, sex, genotyping batch effects, and the first 40 principal components to account for population structure. We identified 95 lead-independent genome-wide significant variants mapped to 129 genes, of which 8 (ALPP, CST5, IL12B, ACVRL1, IL7R, SCGB1A1, NCAM1 and ICAM5) encoded one of the selected proteins in the pSIN score. Of the 95 significant lead-independent variants, 34 were cis-pQTLs mapping to 16 genes, 8 of which encoded proteins included in the pSIN model. Six cis-pQTLs were annotated for multiple closely located genes simultaneously (e.g. rs901886 and rs35929247 were cis-pQTLs for ICAM1, ICAM3, ICAM4 and ICAM5). Additionally, 7 variants were identified as trans-pQTLs mapping to 366 genes, 21 of which encoded proteins selected in the pSIN model. Among these, rs2519093 showed the broadest mapping, acting as a trans-pQTL for 358 genes and a cis-pQTL for two genes (ABO and DBH). Of the 129 genes associated with pSIN, 10 genes were previously identified as GWAS smoking loci[9,21], and 54 genes were previously found in epigenetic studies[10,14,22], 75 (58%) were novel and have previously been reported *to be associated with* body mass index (BMI), diabetes, cancer development, and immunological/haematological traits including lymphocyte counts, eosinophil counts, and white blood cell counts (Supplementary Fig. 3 and Supplementary Data 6–9). Importantly, haematological measurements also showed strong associations with pSIN in the UKB before and after adjusting for smoking status (Supplementary Fig. 4A, B). Genetic correlations using LD score regression (LDSC) analysis highlighted strong genetic correlations of pSIN with current smoking ($r = 0.78$, $p = 1.08 \times 10^{-98}$), never smoking ($r = -0.65$, $p = 1.05 \times 10^{-59}$), maternal smoking around birth ($r = 0.66$, $p = 2.08 \times 10^{-42}$), cannabis use ($r = 0.51$, $p = 1.50 \times 10^{-27}$), smoking-related lung disorders (lung cancer ($r = 0.71$, $p = 1.14 \times 10^{-06}$), COPD ($r = 0.54$, $p = 1.04 \times 10^{-31}$), depression ($r = 0.28$, $p = 9.46 \times 10^{-16}$), ADHD ($r = 0.47$, $p = 8.26 \times 10^{-16}$), multi-site chronic pain ($r = 0.31$, $p = 7.31 \times 10^{-16}$), obesity and fat distribution (BMI $r = 0.33$, $p = 1.832 \times 10^{-22}$), % of leg body fat ($r = 0.33$, $p = 1.05 \times 10^{-23}$) and diabetes ($r = 0.27$, $p = 4.08 \times 10^{-10}$) among others (Supplementary Data 10).

We then studied the relation of pSIN with all available environmental exposures in the UKB (i.e. the exposome). The exposome-wide analysis was conducted using generalised linear models, where each factor was tested separately for its association with pSIN. Models were adjusted for recruitment centre, ethnicity and smoking status. Favourable socio-economic indicators and healthier lifestyle choices, including better housing conditions, lower Townsend deprivation index, higher household income, higher levels of education, higher consumption of fruit and fibre intake, higher levels of physical activity and social interactions were associated with lower levels of pSIN (Supplementary Data 11). Conversely, unhealthy lifestyle choices, poor environment and mood disorders including high salt intake, high consumption of red/processed meat and coffee, alcohol consumption, exposure to air pollutants (PM10, PM2.5, NO2 and NO), maternal

smoking during pregnancy, evening chronotype, sleeping greater than 9 h or less than 7 h or feeling of tiredness or low mood were associated with higher pSIN scores. These findings suggest that beyond smoking behaviour itself, a range of socio-economic, dietary, environmental and behavioural factors each contributed independently to smoking-related damage as reflected by effects on pSIN.

We next explored how different clinical biomarkers and risk factors that were read-outs of the exposome (e.g. obesity as a read-out of diet and physical activity) and those indicative of overall health status at baseline (e.g. lipid levels, creatinine and blood pressure) influenced the pSIN. Firstly, blood biochemistry profiles revealed significant correlations with pSIN. These included markers of inflammation, such as glycA (glycoprotein acetyl, antichymotrypsin) and C-reactive protein, indicators of glucose metabolism such as higher HbA1c levels and biomarkers related to kidney and liver function. In addition, significant associations were observed with sex hormones, lower vitamin D levels and ageing-related biomarkers like telomere length and insulin-like growth factor 1 (Supplementary Fig. 5a and Supplementary Data 12). Additional analyses of decline in clinical function indicated that self-rated health status, older facial ageing, but also clinically assessed ones, including systolic blood pressure, heel bone density, fluid intelligence and lung function, were positively correlated with pSIN (Supplementary Fig. 5b and Supplementary Data 12). Moreover, baseline disease, including type 2 diabetes and arterial stiffness, was strongly correlated with higher pSIN scores (Supplementary Data 12). The findings of this study highlight the substantial and diverse impact of smoking on physiological systems, as reflected by the associations of pSIN with a wide array of clinical biomarkers and health indicators at baseline. We conducted additional sensitivity analyses where the associations between pSIN and clinical biomarkers were further adjusted for smoking status (Supplementary Fig. 6a, b). We observed that while the effect sizes of biomarkers such as HbA1c, GlycA and Triglycerides were attenuated, they remained statistically significant. However, biomarkers such as APOA, HDL cholesterol and Creatinine became non-significant after this adjustment. Similarly, the association between pSIN and clinical risk factors, such as poor self-rated health, was weakened when adjusting for smoking status, while the associations with other risk factors were maintained at a similar level.

To explore the relative contributions of the genome, smoking history and exposome to pSIN, we used a gradient boosting model. We first regressed GWAS significant SNPs against pSIN and calculated the variance explained (Fig. 2d). In our analysis, genetic factors explained a larger proportion of variance in pSIN never smokers (2.7%) and previous smokers (1.6%) compared with current smokers (0.3%). However, overall, genetic factors had a minimal contribution to pSIN. We next estimated the contribution of the exposome, starting with the contribution of smoking history, then added sociodemographic and lifestyle factors and finally added clinical biomarkers and risk factors to the models, calculating the additional proportion of variance explained by each factor (Fig. 2d). Smoking history accounted for the largest proportion of pSIN variance in both the overall population (65.8%), current (52.1%) and previous smokers (16.4%); passive smoking explained 0.2% of the variance in never smokers. Socio-demographic and lifestyle factors contributed most of the variance in never smokers (4.8%), with factors such as air pollution, diet and alcohol consumption making a substantial contribution to levels of pSIN in this subgroup. Lastly, clinical biomarkers and general health indicators provided a comparable amount of additional information across current (2.0%), previous (1.9%) and never (2.0%) smokers.

We further conducted an analysis using multivariate linear models with ANOVA to assess the partial $R^2$ explained by each category when all variables were included in a single model. In the overall UKB population, smoking history accounted for the largest proportion of variance (partial $R^2 = 42.5\%$), followed by clinical biomarkers (5.3%). In contrast, prevalent disease status and polygenic risk scores explained

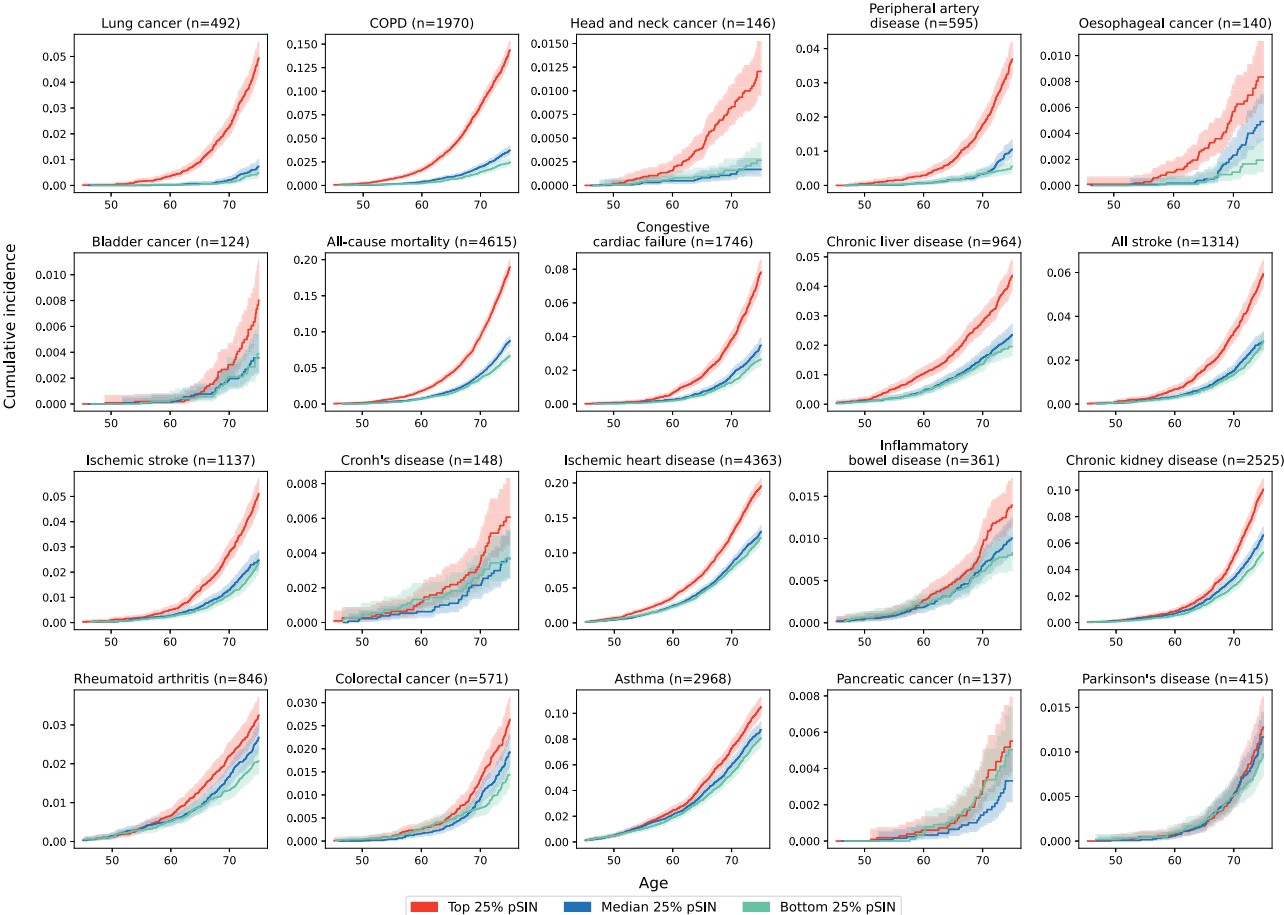

**Fig. 3 | Quartiles of pSIN lead to strongly diverging cumulative incidence of major incident diseases and mortality.** Cumulative incidence plot of top, median and bottom 25% of the pSIN in the whole UKB population with 95% confidence interval shown as lighter shading. X-axis denotes the chronological age and Y-axis denotes the cumulative incidence. Cumulative incidence and number at risk at each age point are shown in Supplementary Data 15 and Supplementary Data 16.

only 0.1 and 0.02% of the variance, respectively. Notably, technical factors—including fasting time, assessment centre and assay batch—contributed just 0.3% of the variance, underscoring the robustness of the model (Supplementary Fig. 7). When stratified by smoking status, clinical biomarkers explained the largest proportion of variance in current smokers (9.9%), followed by smoking history variables other than smoking status (7.8%) and lifestyle factors (3.7%). Clinical biomarkers also remained important contributors to pSIN in previous smokers (4.1%) and never smokers (4.2%). Among previous smokers, smoking history explained the greatest variance (6.9%), whereas in never smokers, it accounted for only 0.03% (passive smoking). Finally, technical factors contributed modestly to pSIN variance in current (1.4%), previous (0.4%) and never smokers (0.3%) (Supplementary Fig. 7).

These findings underscore the multifactorial nature of pSIN; it is predominantly determined by smoking history in the population overall and in current and previous smokers, but also the genome and exposome are related to pSIN, even in non-smokers.

**Associations of pSIN with risk of morbidity and mortality**

Over a mean follow-up time of 13.3 years (SD = 2.2) in the UKB, 4615 deaths were observed. We tested the association of pSIN with 27 major disease outcomes and all-cause mortality. In the overall UKB population, pSIN was significantly associated (FDR < 0.05) with 19/28 outcomes independent of the confounding factors. These included lung cancer (HR = 1.97, CI:1.83, 2.11), COPD (1.72, 1.67, 1.78) and head and

neck cancer (1.64, 1.44, 1.86), ranking as the top 3 based on their hazard ratios per SD increase in pSIN. pSIN was able to differentiate subtypes of inflammatory bowel disease, with pSIN associating with higher risks in Crohn's disease ($p = 2.34 \times 10^{-2}$) but not ulcerative colitis ($p = 5.99 \times 10^{-2}$), concordant with previous literature[23]. Interestingly, pSIN displayed a protective effect for Parkinson's disease (0.84, 0.75, 0.94) (Supplementary Fig. 8 and Supplementary Data 13), one of the few common disorders with a lower risk in smokers[24]. pSIN was also significantly associated with mortality (1.32, 1.28, 1.36). Associations of individual proteins with incident health outcomes are shown in Supplementary Fig. 9. We further validated the association with lung cancer, COPD, any vascular disease, any respiratory disease, ischemic stroke, all stroke, ischemic heart disease and mortality in the CKB, all of which stayed significant after adjusting for confounding factors (Supplementary Data 14).

For mortality and the 18 diseases that were significantly associated with pSIN, we further tested if participants in the top, median and bottom quartiles of the pSIN exhibited divergent cumulative incidence in each outcome (Fig. 3). We found that 11 diseases showed more than 2-fold higher cumulative incidence when comparing the top and bottom quartiles of pSIN at age 75 years including lung cancer, peripheral artery disease, COPD, head and neck cancer, esophageal cancer, congestive cardiac failure, all-cause mortality, chronic liver disease, ischemic stroke, all stroke and bladder cancer. The cumulative incidence and number of risks in each age group are shown in Supplementary Data 15 and 16.

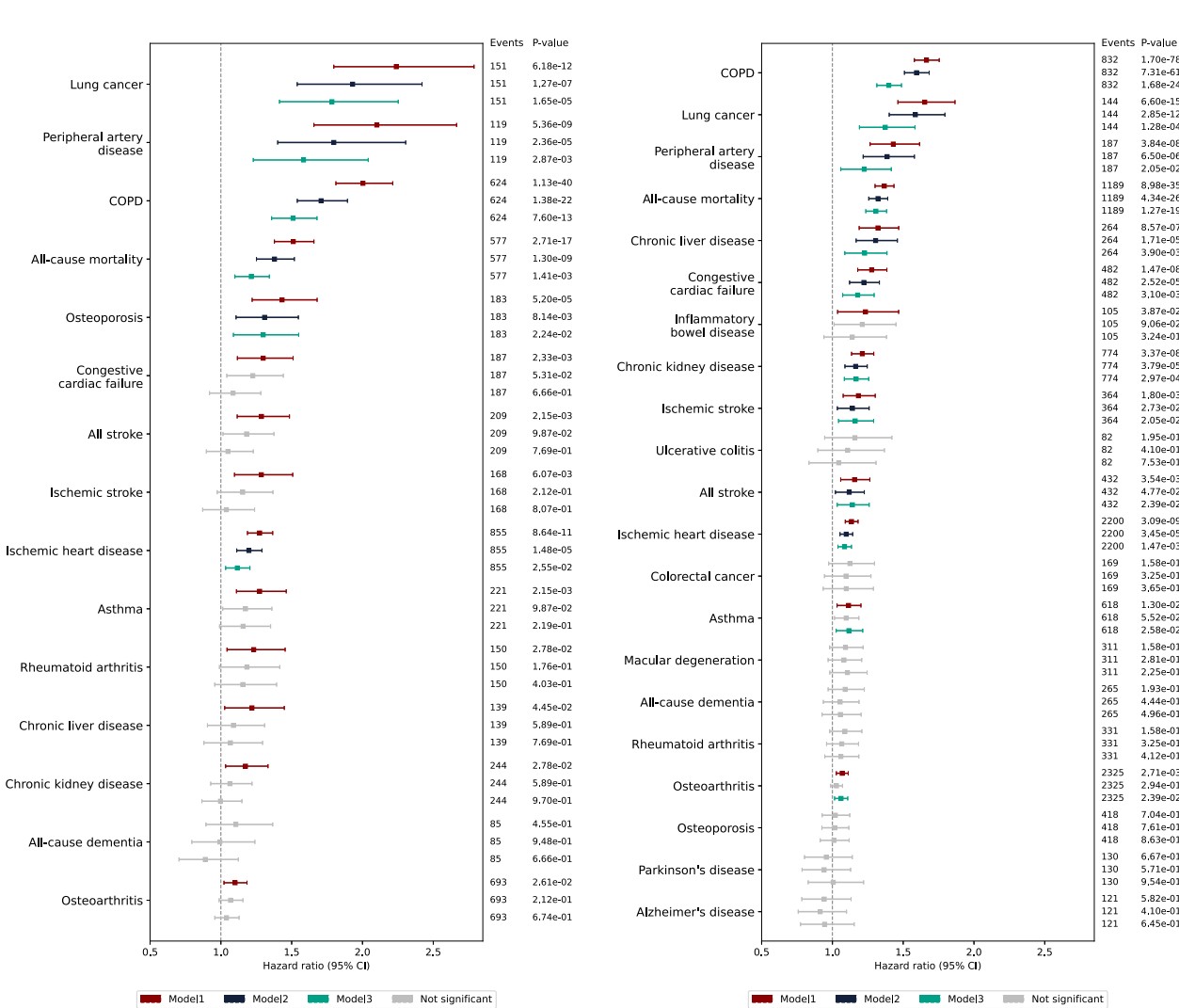

**Fig. 4 | pSIN differentiates future risks of morbidities and mortality in current and previous smokers. A** shows association between pSIN and health outcomes which has at least 80 cases during follow-up time using multi-variate cox proportional hazard model in current smokers. In model 1, the exposure was pSIN without adjusting for any covariate as age and sex already been regressed out in protein level. Model 2 was adjusted for recruitment centre, Townsend deprivation index, IPAQ physical activity group, ethnicity, alcohol frequency, BMI and education years. Model 3 further adjusted for smoking pack years and smoking cessation time. Hazard ratios with 95% confidence intervals were shown. *P*-values were corrected for FDR multiple testing and non-significant associations after corrections were shown as grey colour. **B** shows association between pSIN and health outcomes in previous smokers.

To evaluate the additional information provided by the pSIN compared to self-reported smoking habits, we analysed their associations with current smokers and previous smokers. In current smokers, pSIN was significantly associated with 6/15 major smoking-related health outcomes (with a more than 80 incident cases cutoff), including lung cancer, peripheral artery disease, COPD, all-cause mortality, osteoporosis and ischemic heart disease after adjusting for lifestyle factors and smoking pack-years (Fig. 4a, Supplementary Data 17). Further, participants in the highest quartile of pSIN had greater than 2-fold higher cumulative incidence at age 75 years compared to those in the lowest quartile for all evaluated outcomes, except for ischemic heart disease, which showed a 1.59-fold higher risk (Supplementary Fig. 10a). Cumulative incidence and number at risk at each age are provided in Supplementary Data 18 and 19.

Among previous smokers, pSIN was significantly associated with higher risks of 12 disease outcomes, including COPD, lung cancer, peripheral artery disease and mortality, among others, after adjusting for pack-years smoked and number of years since cessation (Fig. 4b and Supplementary Data 20). Participants in the highest quartile of pSIN had greater than 2-fold higher cumulative incidence at age 75 years compared to those in the lowest quartile in 2 of the morbidities, including lung cancer (3.43-fold higher risk) and COPD (2.06-fold higher risk). The cumulative incidence associated with pSIN for each disease is shown in Supplementary Fig. 10b and Supplementary Data 21, 22.

## Use of pSIN to differentiate the recovery status of previous smokers

Analyses of changes in mean pSIN levels by years since smoking cessation among previous smokers indicated that it took only 2 years for the pSIN to decline to a threshold that differentiated current from never smokers at a false positive rate (FPR) of 0.05 (Supplementary Fig. 11). However, the variation of pSIN in each individual with the same cessation years is large and for a substantial number of people, the

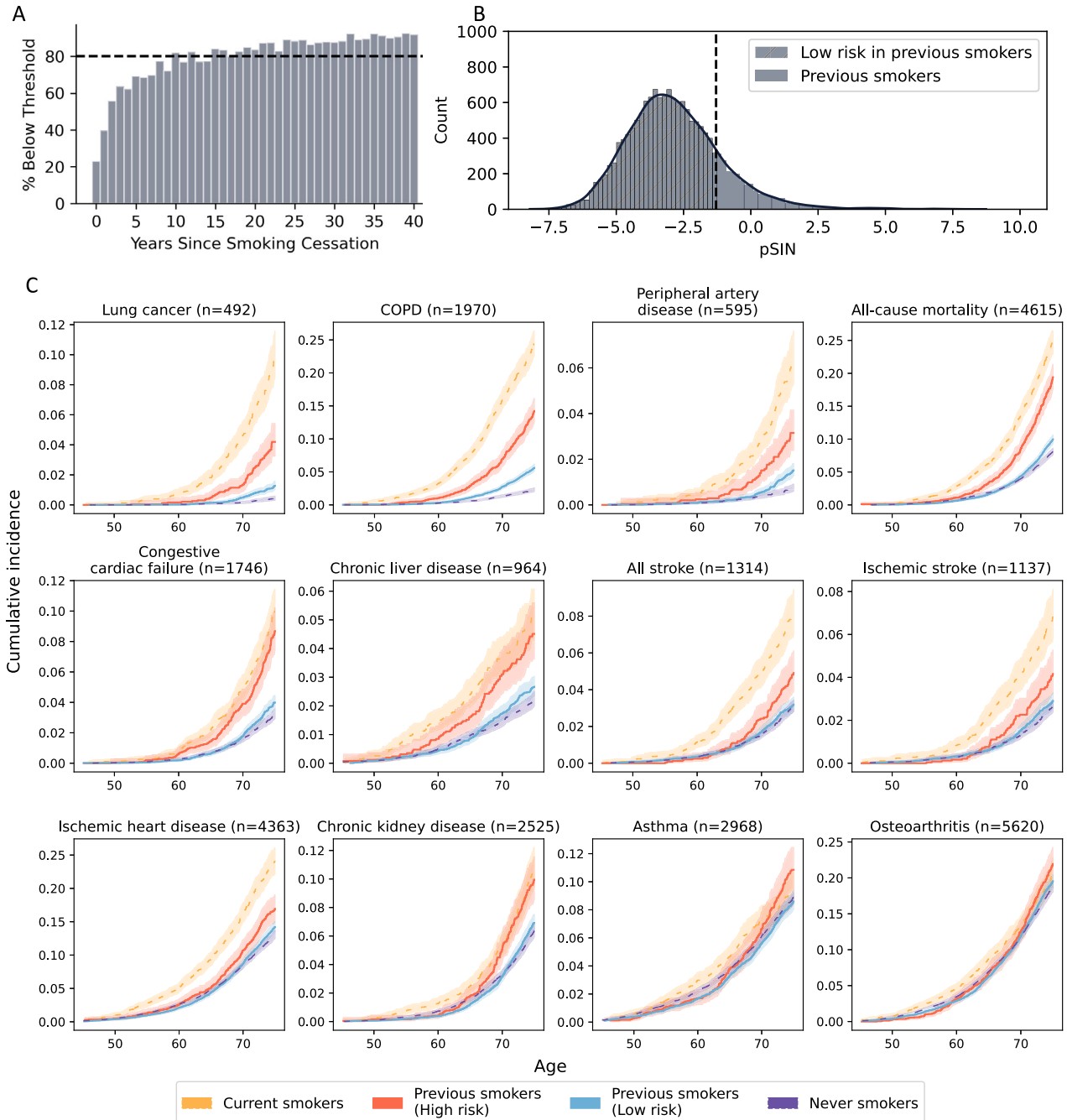

**Fig. 5 | Previous smokers with pSIN similar to never smokers (recovery) show lower morbidity and mortality risks. A** shows the percentage of the previous smoker population in each year of smoking cessation bin who have a pSIN lower than the cutoff value used to differentiate current smokers. This percentage first reaches 80% after 10 years of smoking cessation. **B** shows the distribution of pSIN in previous smokers. Dotted line denotes the cut-off when differentiating current smokers from never smokers at FPR of 0.05, dividing previous smokers into two groups. Hashed part denotes the group of previous smokers with a similar pSIN as never smokers (recovery). **C** shows a cumulative incidence plot of low and high-risk groups defined by pSIN in previous smokers (orange and blue) with self-reported current smokers as positive control (yellow) and self-reported never smokers as negative control (purple). A 95% confidence interval was shown in shaded area. Cumulative incidence and number at risk charts are shown in Supplementary Data 24 and 25, respectively.

pSIN remains high despite quitting smoking. Indeed, at least 10 years were needed for more than 80% of the previous smokers to have a pSIN below the threshold, consistent with findings in the British Doctors study[25] (Fig. 5a). Additionally, the mean levels of pSIN of previous smokers plateaued, approaching the levels seen in never smokers after approximately 25 years after cessation of smoking (Supplementary Fig. 11). Using this cut-off, we identified two clinically relevant subgroups among previous smokers: (1) those whose proteomic profile

was similar to that of the current smokers and (2) those whose proteomic profiles were similar to the never smokers—henceforth referred to as the 'recovered' group (Fig. 5b; $n = 2576/15{,}404$).

To determine whether this differentiation between these two groups of previous smokers was clinically relevant, we evaluated the disease and mortality risk differences between the two groups, adjusting for lifestyle factors, smoking pack-years and smoking cessation years. Within the group of previous smokers, the recovered

group had significantly lower risks for 10 of the disease outcomes, including COPD (HR = 0.33; CI: 0.29, 0.38), lung cancer (0.37, 0.27, 0.52) and all-cause mortality (0.52, 0.46, 0.59). (Supplementary Fig. 12, Supplementary Data 23). Importantly, for most diseases, cumulative lifetime risks for recovered groups were comparable to those for never smokers (Fig. 5c).

In contrast, previous smokers who had not recovered from smoking damage exhibited lifetime cumulative incidences of asthma ($p = 0.72$), chronic kidney disease ($p = 0.62$), chronic liver disease ($p = 0.13$) and congestive cardiac failure ($p = 0.12$) that were comparable to those of current smokers. However, for most respiratory and cardiovascular diseases, their risk was significantly lower than that of current smokers. Notably, diseases such as lung cancer and peripheral artery disease showed more than a 50% reduction in cumulative incidence by age 75 (Supplementary Data 24 and 25).

In addition, we also identified current smokers whose pSIN scores were similar to those of never smokers (Supplementary Fig. 13a, $n = 737/4732$). After adjusting for lifestyle factors and smoking pack-years, this population had significantly lower risks for peripheral artery disease (HR = 0.13, CI: 0.03,0.53), lung cancer (0.15, 0.05, 0.46) and COPD (0.25, 0.16, 0.39) in addition to all-cause mortality (0.53, 0.38, 0.73) (Supplementary Fig. 13b, Supplementary Data 26). Supplementary Fig. 13c shows that when current smokers had similar pSIN to never smokers, they exhibited cumulative disease risks comparable to those of never smokers for all tested health-related outcomes with the exception of COPD (risk at 70 years: 2.5%, $p = 9.5 \times 10^{-5}$) and all-cause mortality (risk at 70: 6.4%, $p = 4.59 \times 10^{-2}$) which were associated with a lower absolute risk compared to high-risk current smokers (Supplementary Data 27 and 28).

## Discussion

Using a machine learning analysis of plasma proteomic data in UKB, we generated the pSIN score, which differentiated current smokers from never smokers with high accuracy and validated this score in CKB. Although the model was trained to classify current smokers vs. never smokers, examining variance explained by smoking history allows us to assess how much of the proteomic signature is attributable to smoking behaviour versus other variables that might influence the biological markers, such as genetics or other environmental factors. We found that pSIN levels in the population and cigarette smokers were determined mainly by smoking history but also by genetic, environmental and clinical risk factors and blood-based biomarkers of morbidity and mortality. Higher levels of pSIN were strongly associated with mortality and 18 out of 27 major diseases and the associations were independent of self-reported smoking exposure in smokers. Among previous smokers, the risks of morbidity and mortality were attenuated compared with current cigarette smokers, but pSIN levels predicted the residual risk of mortality and morbidity.

The findings from the present study corroborate and expand upon previous research studies, particularly highlighting associations of smoking with COPD, peripheral artery disease and lung cancer[26,27]. Comparisons with the available worldwide evidence on this topic indicated that 36 of 51 smoking-associated proteins identified in the present study were independently associated with smoking in genome[9,21,28], epigenome[10,14,22], transcriptome[15,29,30] (Supplementary Fig. 14) or individual protein-based studies (Supplementary Data 29). Most of the top 20 proteins contributing to pSIN align with these findings, with novel associations identified for the first time, such as DKK4, SLITRK1, KLK13 and CA4. Assessing the predictive performance of our pSIN model against previous epigenomic studies as the gold standard, we achieved comparable discrimination power for smoking status. For instance, Sugden et al.[9] achieved an AUC of 0.93 and 0.81 based on 2623 CpGs; Bollepalli et al.[14] achieved an average sensitivity of 0.81 with a specificity of 0.85 when using 121 CpGs; and Maas et al.[22] achieved a 5-fold CV AUC of 0.897 using 13 CpGs. In contrast, our pSIN

model demonstrated a robust performance with a 5-fold cross-validation AUC of 0.96 within the training set and an AUC of 0.95 in the test dataset. Furthermore, for identifying current smokers with high specificity, pSIN outperformed existing models with a sensitivity of 0.85 and a specificity of 0.95. We find that pSIN is genetically correlated with maternal and passive smoking. However, in never smokers, their contribution to the pSIN levels is small (0.2%). Interestingly, beyond matched performance, we also observed a similar distribution of the DNA methylation smoking score built by Elliott et al.[31] in current, previous and never smokers (Supplementary Fig. 15) compared with pSIN in UKB.

Beyond the new proteins associated with smoking and the high levels of accuracy of the models, the major novelty of the present study was the link between the composite effects of these proteins and with individual risk of morbidity and mortality. Previous omics studies, which have been limited by sample size or short duration of follow-up and primarily focused on genetic correlations with diseases, were unable to fully assess associations of selected omics markers with incident disease outcomes. In contrast, the present study provided a more comprehensive analysis, which included an investigation of associations of pSIN with 27 major incident diseases and with all-cause mortality. We demonstrated that pSIN effectively differentiated risks of disease outcomes in the general population and provided additional predictive value for assessing risks of morbidity and mortality among both current and previous smokers, independent of confounding factors and smoking history. The present study highlights the clinical relevance of pSIN for predicting disease risks, providing an objective, personalised measure of smoking history that can be translated into a risk estimate of various diseases and death. For example, previous smokers who have similar pSIN to never smokers are found to have significantly reduced incident morbidity and mortality risks, potentially comparable with those of never smokers. This suggests that pSIN can serve as an objective test to assess the magnitude of recovery from smoking-related damage among previous smokers. We also demonstrated that for diseases like lung cancer and peripheral artery disease, the lifetime risks declined by more than 50% immediately after smoking cessation, even for the highest-risk group of previous smokers compared to current smokers. Importantly, we found an inverse association between pSIN and Parkinson's disease. The reliability of this inverse association was demonstrated in the 60-year follow-up of the British Doctors' Study[32] with molecular evidence suggesting that nicotine and related chemicals associated with smoking may reduce MPTP-induced dopaminergic toxicity or inhibit the enzymatic oxidation of dopamine[33,34].

Comprehensive analyses of omics and questionnaire data in UKB enabled us to explore the associations of genetic and environmental factors associated with pSIN. Comparisons of current smoking with behaviour and environmental factors indicated that the impact of the genome on pSIN was limited in smokers and previous smokers, and the related genetic correlation analysis indicated a substantial overlap in the genes associated with pSIN and smoking habits. This suggests most of the disease risks associated with smoking are modifiable and preventable. Importantly, the findings that some participants self-reported as never smokers exhibited pSIN levels comparable with those of current smokers have potentially important health relevance. We hypothesise that this group may include individuals who did not accurately report their smoking status, perhaps reflecting a 'social desirability' bias[5]. Indeed, in CKB, the self-reported smoking history was validated by measurements of exhaled CO among all participants at baseline; 20.3% of this population who reported as never smokers had a CO level exceeding 10 ppm, a widely recognised threshold to indicate smoking[8]. Alternatively, they may be exposed to other factors that are associated with the same protein pathology as pSIN. Based on the genetic correlation, BMI emerged as a risk factor for diseases that share a common pathogenesis with smoking, potentially involving

processes such as epithelial cell proliferation, regulation of the immune system, cell growth and T-cell activation. Additionally, our study demonstrated robust associations between pSIN and maternal smoking, supported by both direct questionnaire-based assessments and genetic correlation analyses. Previous reports have consistently demonstrated that cigarette smoking during pregnancy has been linked with disease outcomes in offspring, including neurodevelopmental and behavioural issues, obesity, hypertension, type 2 diabetes and impaired lung function[35] as well as epigenetic changes[36]. In addition, the findings of the present study provide long-term molecular evidence of these effects by assessment of pSIN. Last but not least, while it is well established that air pollution, especially fine particles, results in higher risks of lung cancer, COPD and cardiovascular diseases[37,38], the present study provides details of biological instruments to quantify the long-term effects of smoking in large-scale blood-based population studies.

Despite being the largest study conducted to date, the present study was constrained by a limited number of proteins compared with the total number of known proteins. Secondly, Olink measurements are relative quantifications suitable for large cohort studies due to high throughput and cost considerations; any translation of such assays into clinical practice would require replication of these findings using absolute quantification. Third, the study was limited only to the subset of UKB where the plasma proteome was measured, limiting the statistical power to detect associations between pSIN and diseases where rarer outcomes could not be evaluated. Further, the causal relationships between environmental exposures and smoking are inherently complex. While our analysis aimed to leverage protein signatures within pSIN to understand the downstream pathology of smoking, it is essential to acknowledge that some associations identified may not be directly attributable to smoking behaviour. Instead, these associations could arise from independent effects of environmental exposures and smoking-related behaviours on proteins that are part of the pSIN. This limitation underscores the challenges of disentangling causal pathways in observational data and highlights the necessity for cautious interpretation when utilising scores based on protein networks to elucidate aetiological mechanisms. Future studies should consider complementary approaches, such as Mendelian randomisation or experimental validation, to disentangle these independent effects and strengthen causal inferences.

We acknowledge that while our study provides a strong foundation for understanding the molecular signatures of smoking and its associations with disease risk, further steps are needed to explore its potential clinical applications. To bridge the gap between discovery and translational research, we propose that future studies should evaluate the feasibility of integrating pSIN into primary care settings. This could involve testing its utility in a clinical trial designed to assess whether proteomic-based risk stratification improves early detection and targeted interventions for high-risk individuals. Key considerations include the cost-effectiveness of measuring the 51 proteomic biomarkers using Olink or alternative platforms, as well as the logistical challenges of incorporating these assessments into routine clinical workflows. Additionally, implementing pSIN in practice would require careful evaluation of whether it provides actionable information beyond traditional risk factors and whether its use could justify the costs associated with proteomic profiling and the collection of comprehensive clinical, epidemiological and environmental data. While high-throughput proteomics is currently expensive, costs are expected to decline with technological advancements, potentially making it a viable tool for personalised risk assessment in the future. As with any predictive model, the accuracy of pSIN depends on the quality of the data and the model's generalisability. Although we have demonstrated high predictive accuracy (AUC = 0.95) in the UKB and CKB datasets, there are potential limitations, particularly when applying the signature to new populations or for specific uses (e.g. insurance assessments). Although we demonstrated that a high pSIN score is strongly associated with smoking-related biological changes, it does not guarantee that someone with a high score has indeed smoked. Other factors, including genetic predisposition, environmental exposures and other health conditions, could influence proteomic profiles. Hence, its use as a definitive marker of smoking history should be approached with caution.

Overall, the present study demonstrated that the blood proteome is a powerful tool to measure downstream molecular changes associated with cigarette smoking and a reliable measure to quantify the risks of smoking-related diseases. This study adds to the available evidence linking molecular signatures of smoking with the risk of common diseases associated with smoking and all-cause mortality and affords an opportunity to inform the hazards of current and previous smoking for the risk of morbidity and mortality in population studies.

## Methods
### Study cohorts
**Ethics approval**. UKB data use (Project Application Number 61054) was approved by the UKB according to their established access procedures. UKB has approval from the North West Multi-centre Research Ethics Committee (MREC) as a Research Tissue Bank (RTB) and as such, researchers using UKB data do not require separate ethical clearance and can operate under the RTB approval. The CKB (Oxford Tropical Research Ethics Committee 025-04) complies with all the required ethical standards for medical research on human subjects. Ethical approvals were granted and have been maintained by the relevant institutional ethical research committees in the UK and China.

**UK Biobank (UKB)**. The UKB is a prospective cohort study which includes 502,505 participants recruited between 2006 and 2010[39]. Cohort population characteristics were summarised in Supplementary Data 1. Smoking behaviours, including smoking status, number of cigarettes per day, smoking start age, age stopped smoking, type of tobacco smoked, exposure to tobacco smoke at home and exposure to tobacco smoke outside, were collected at baseline using touchscreen-based questionnaires. Participants were classified as current smokers, previous smokers and never smokers. We classified 'occasional smokers' as never smokers to ensure consistency in our comparison groups and to align with prior epidemiological studies. Occasional smokers comprised a very small proportion of the population and did not report a sustained smoking history. Given that our study focuses on the long-term molecular impact of regular smoking, we categorised occasional smokers with never smokers to avoid potential misclassification bias and to maintain a clear distinction between individuals with significant smoking exposure and those without. Smoking pack-years were calculated as the Number of cigarettes per day/ 20 × (Age stopped smoking – Age started smoking). Passive exposure was defined as a binary variable, indicating whether an individual had been exposed to tobacco smoking either at home or outside. UKB blood biomarkers were measured using the non-fasting blood serum samples collected at baseline. Sample processing, quality control and technical variation adjustment steps were performed by UKB before data release and were previously described[40]. Sociodemographic, lifestyle and environment, early life factors, family history, psychosocial factors and health and medical history collected through touchscreen-based questionnaires at baseline were used as exposures for exposome-wide association studies. 176 variables common to both males and females were used in the analysis. Continuous exposures were centred and standardised, except for age. Ordinal categorical exposures were recoded as ordered categorical variables. Nominal categorical exposures were analysed with the most common category set as the reference and dummy variables were generated accordingly. Details of the variables and processing methods were described elsewhere[41].

Missing data was imputed using a random-forest-based algorithm provided by the R package missRanger[42] when used as a covariate in linear association models (Townsend deprivation index, IPAQ physical activity group, ethnicity, alcohol frequency, BMI and education years). Recruitment centre has a missingness of 0.00%, ethnicity has a missingness of 0.55%, alcohol frequency has a missingness of 0.30%, BMI has a missingness of 0.62%, IPQA activity group has a missingness of 19.93%, Townsend deprivation index has a missingness of 0.12% and education years has a missingness of 2.02%. Imputation was performed with default hyperparameters using a maximum of 10 iterations and 200 trees. Linked hospital inpatient data, primary care data and cancer register data were accessed from the UKB data portal on May 23, 2023, with a censoring date of Oct 31, 2022, Jul 31, 2021, Feb 28, 2018, for participants recruited in England, Scotland and Wales, respectively. The follow-up time is between 8 and 16 years. Mortality data and cause of death information were accessed from the UKB data portal on May 23, 2023, with a censoring date of November 30, 2022. The follow-up time is between 12 and 16 years. Methods and ICD diagnosis codes used to identify prevalent and incident chronic disease in UKB are shown in Supplementary Data 30.

**Proteomics assessment.** Proteomic assessments in the UKB were performed using the Olink Explore 3072 platform, where the biomarker library was determined by pre-existing knowledge of pathways involved in cardiometabolic, inflammation, neurology and oncology. After excluding participants with more than 20% missingness in protein measurements or those with smoking status information missing or answered 'prefer not to say' at the baseline assessment and removing the non-randomised samples (batches 0–7), 43,914 participants were retained for the analyses. The randomised participants selected for proteomic profiling were previously shown to be representative of the original UKB full cohort[43]. Proteomics data were provided in Normalised Protein eXpression (NPX) values on a log2 scale, where quality control[44] and a 2-step approach of within-batches and cross-batches intensity normalisation[45] was performed pre-releasing the data. Proteins missing in over 20% of the sample were excluded from the study (NPM1, PCOLCE, CTSS, GLIPR1), leaving a total of 2917 proteins for the analysis. After regressing out recruitment age and sex, the proteomic data were further processed by first rescaling to the value between 0 and 1 and then centring on the median. To address the concern regarding the potential residual signal of age and sex in the proteomic expression data after linear regression, we performed additional analyses to rigorously evaluate the effectiveness of the regression process. Specifically, we trained gradient boosting models to assess whether any predictive signal for age or sex remained in the regressed data. For age, we trained regression models to predict age using each protein in the regression dataset. The results showed a mean $R^2$ of −0.0042 (SD = 0.0089), indicating that the models performed no better than random chance. This suggests that the linear regression successfully removed age-related signals from the proteomic data. For sex, we trained classification models to predict sex using each protein in the regressed dataset. The models achieved a mean AUC of 0.51 (SD = 0.017), which is equivalent to random guessing. This confirms that sex-related signals were effectively removed by the regression process.

**Genomics assessment.** UKB genotyping was conducted by Affymetrix using a bespoke BiLEVE Axiom array for ~50k participants and the remaining ~ 450k on the Affymetrix UKB Axiom array. As the two arrays are broadly comparable, with over 95% overlap in assessed gene variants, they were combined. All genetic data were quality-controlled and imputed by the UKB. Detailed information on the genotyping process and technical methods is available online[46]. Genetic data were phased prior to imputation with SHAPEIT3, followed by imputations using IMPUTE2. Details on genetic imputations are provided elsewhere[47].

**China Kadoorie Biobank.** The CKB is a prospective cohort study which includes 512,724 participants recruited from 10 geographically diverse regions between 2004 and 2008. Smoking variables were collected by laptop-based questionnaires at baseline. Smoking status was collected as never smokers, occasional smokers, ex-regular smokers and smokers. We then redefine never smokers and occasional smokers as never smokers, ex-regular smokers as previous smokers and smokers as current smokers. This study restricted the analysis to participants where the proteomic profile was measured with the Olink platform ($n = 3977$). Details of the study design and methods were previously described[48].

**Proteomics assessment.** Proteomic assessments in CKB were performed using the Olink Explore 3072 platform. Plasma samples from selected participants were sub-aliquoted to 2 batches and were measured by the laboratory at Uppsala, Sweden (batch 1, 1463 proteins) and Boston, USA (batch 2, 1460 proteins) separately. The CKB Olink data were provided in NPX values on the log2 scale, where the CKB team has performed quality control, a 2-step approach of internal control and inter-plate control. Any protein was excluded if it was not available in either UKB or CKB (HLA_A, ERVV_1, CD97, FGFR10P, LRMP, CASC4, ADRS, HARS, WISP2, FOPNL, WISP1). After regressing out age and sex in the CKB cohort, proteomic data were further processed by first rescaling to the value between 0 and 1 and then centering on the median.

### Statistical analysis

A descriptive analysis of population characteristics was performed using the R package CBCgrps[49]. The study design and analysis pipeline are illustrated in Fig. 1.

**Proteomic signatures of smoking.** Proteomic profiles of smoking were constructed by comparing current smokers with never smokers, excluding passive smokers, using gradient boosting. Samples were randomly split into 70% training ($n = 13,343$ for never smokers, $n = 3312$ for current smokers) and 30% testing dataset ($n = 5719$ for never smokers, $n = 1420$ for current smokers). Within the training dataset, a gradient boosting machine learning model, including all 2911 proteins, was trained to differentiate never smokers and current smokers. The model was first hyperparameter-tuned using a Tree-structured Parzen Estimator-based method provided by the Optuna package in Python. Hyperparameters within a pre-set range were searched and optimised across 200 trials to maximise the 5-fold cross-validated ROC AUC score. After hyperparameter tuning, the performance of the best parameter in the training dataset with 5-fold cross-validation and in a 30% left-out testing dataset was assessed.

**Feature interpretation and selection.** To characterise the feature importance, SHapley Additive exPlanation (SHAP), a local tree explaining method based on game theory, was used[50]. SHAP calculates the contribution of each feature to the outcome in each participant and extends these local explanations to also capture interactions between features directly. Compared to traditionally used permutation feature importance, SHAP plots can display the magnitude, prevalence and direction of a feature's effect. We then used a SHAP-based Boruta selection method provided by the shap-hypetune package[51] to select all relevant features contributing to smoking status prediction. The Boruta algorithm enhances feature selection by creating randomly permuted shadow features that serve as a baseline for comparison. Specifically, shadow features are duplicates of the original features with their values randomly shuffled to break any associations with the target variable. The algorithm compares the mean absolute SHAP values of the original features against those of the shadow features. Features are retained only if they demonstrate a higher mean SHAP value than the highest-scoring shadow feature, thereby ensuring their predictive importance is not due to random chance.

In our study, the algorithm was executed for 200 iterations to ensure robustness and stability in feature ranking. Features falling within the bottom 5% of importance scores (tail 5%) were systematically rejected as they were unlikely to contribute meaningfully to the model's performance. Following the feature selection process, the refined model—consisting of Boruta-selected features—underwent another round of hyperparameter tuning to optimise its performance before further analysis. All model tuning and feature selection steps were performed within the 70% training set in UKB.

**Proteomic Smoking Index (pSIN).** The pSIN for the full UKB sample ($n = 43,914$) was calculated using a robust methodology to mitigate the risk of overfitting. This process involved employing 5-fold cross-validation to ensure the reliability of the results. After identifying the best hyperparameters and selecting the proteins using the Boruta method, a gradient boosting model was trained within each fold. Subsequently, the predicted raw score for the corresponding test set was generated. For binary classification tasks, this raw score corresponds to the log odds of the positive class (in this case, being a current smoker). The LightGBM model typically outputs raw scores (logits) in the range of approximately −10 to 10, where a score of 0 indicates a neutral prediction, corresponding to a 50% probability of being in the positive class (sigmoid (0) = 0.5). Scores closer to 10 indicate a high confidence prediction for the positive class (sigmoid (10)≈0.99995 or ~99.995% probability), while scores closer to −10 indicate a high confidence prediction for the negative class (sigmoid (−10)≈4.5 × 10$^{-5}$ or ~0.0045% probability). In our analysis, we set the classification threshold at a raw score of −1.29, which corresponds to the point where the FPR is 0.05. This threshold was chosen to balance sensitivity and specificity in our predictions. pSIN higher than the threshold indicates a higher likelihood of being in the positive class (smoker), with larger values reflecting greater confidence, while pSIN smaller than the threshold indicates a higher likelihood of being in the negative class (non-smoker), with more negative values reflecting greater confidence. These predicted raw scores from the test sets of each fold were then aggregated to create a comprehensive measure of smoking protein profiles for the entire population. This approach allowed for a more robust estimation of the impact of smoking on protein profiles across the UKB cohort compared to using only one model trained on training data to calculate pSIN for the entire population. External validation in CKB was performed to further test the possibility of the overfitting problem. For external validation, a model with the optimised hyperparameter was trained in the UKB training dataset and was tested in the CKB. The performances of identifying current smokers from never smokers were compared.

**Function annotation for proteins.** Individual protein function annotation (GO: molecular functions) was extracted from the GO database using clusterProfiler v.4.2.2 in R[52]. No enrichment analysis was performed. Tissue-specific protein expression data were extracted from the GTEx project[16] database v.8 and the heatmap was plotted using the FUMAGWAS webtool. Values shown on the heatmap were the average of normalised expression per gene. Differential expression genes (DEG) were identified by a 2-sided t-test per tissue type versus all other tissue types. Genes with a Bonferroni corrected $p < 0.05$ and absolute log fold change >0.58 were selected as DEG and were shown in red colour.

**Association of smoking history, clinical biomarkers and risk factors, haematological measurements, exposome-wide association analysis with pSIN.** To test the association between self-reported smoking habits, social-demographic factors, lifestyle factors, blood biochemistry biomarkers and clinical risk factors with pSIN, generalised linear models from the statsmodel v.0.14.0 package[53] were used.

For associations between smoking history and pSIN, models were adjusted for basic socioeconomic factors, including recruitment centre, ethnicity, education years and the Townsend deprivation index. Each smoking-related exposure was considered separately. Similarly, smoking cessation years were analysed separately as either continuous or categorical variables in distinct models, with categorical analyses designed to explore dose-response relationships.

For associations between blood biomarkers/clinical risk factors/haematological measurements to pSIN, continuous exposure variables, standardisation was applied before inclusion in the models. Associations were adjusted additionally for lifestyle factors, including IPAQ activity group and alcohol intake frequency, as they are known to confound physiological status and liver damage. To explore the added value of pSIN compared to self-reported smoking status, a sensitivity analysis was performed, adjusting additionally for self-reported smoking status.

For exposome-wide association analysis (all available socio-economic and lifestyle variables available in UKB), models were only adjusted for the most basic recruitment centre, ethnicity and smoking status to allow exploration of a wide range of potential associations between the exposome and pSIN without masking potential signals by controlling for too many variables.

Social-economic and lifestyle confounding factors were chosen based on previous literatures[54–56].

P-values resulting from these analyses were corrected for FDR multiple testing.

**Identification of genes influencing pSIN.** GWAS was conducted using SAIGE software V1.09. For constructing a genetic relationship matrix (GRM) in step 1, we used the pruned genotype dataset. Genotype pruning was conducted in PLINK software using the 'indep-pairwise' option with an $r^2$ of 0.5, a window size of 1000 markers and a step size of 100 markers. We further used the 'LOCO = TRUE' option to construct the GRM. The GWAS analyses were adjusted for age, sex, batch effects and 40 gene principal components identified within UKB genotyping data[47]. Gene mapping was performed using the FUMAGWAS web tool, where the maximum p-value for lead SNP was set as $5 × 10^{-8}$ and the maximum distance to genes was set as 10 kb. LDSC analysis was performed using online tools Complex-Traits Genetics Visual Labs (CTG-VL)[57] where summary statistics of GWAS against pSIN were correlated to publicly available GWAS results of 1461 traits in the database. Significant correlations were identified if the FDR-adjusted $p$-value was smaller than 0.05.

**Calculating the contribution of each category to pSIN.** To investigate how much of the additional variance explained does each category of genetics, smoking history, social-demographic and lifestyle and clinical biomarkers and risk factors have on pSIN, we constructed four gradient boosting models. These models were sequentially augmented by adding each category, starting with genetics, followed by smoking history, then social-demographic and lifestyle factors and finally clinical biomarkers and risk factors. Each model was initially trained on the 70% training dataset and subsequently, the variance explained was assessed using the 30% testing dataset using the scikit-learn package[58]. The additional variance explained by each category was determined by subtracting the variance explained by the model that contains the information on this category from that of the previous model that did not include this category. The same procedure was carried out for the whole population, but also in current smokers, previous smokers and never smokers to investigate how each category contributes to pSIN differently among the smoking status-stratified population. The additional variance explained was then plotted in a stacked bar chart for comparison.

Further, a multivariate linear regression model, followed by ANOVA using statsmodel[53] was used to estimate the relative $R^2$ for each variable. Sum of relative $R^2$ of all variables from the same category was reported. In addition to the previous analysis, prevalent diseases of

lung cancer and COPD, Polygenic Risk Scores (PRS) of smoking initiation, lung cancer and COPD, technical measurements including fasting time, OLINK batches and assessment centre, anthropological measures including standing height, BMI, weight, hip and waist circumference were also included in the model. Dietary factors were included in the lifestyle factors. For PRS calculation, we identified recent PRS from the Polygenic Score (PGS) catalogue, selecting scores derived in predominantly European populations that did not overlap with the UKB cohort. We calculated these PRS as weighted sums ∑(no. risk alleles × effect size) in the UKB v3 imputed genotype data. Details of the method were described elsewhere[59]. PGS catalogue entries used to calculate PRS were as follows: Smoking initiation: PGS003360[28], Lung cancer: PGS000078[60] and COPD: PGS001332[61].

**Associating pSIN with future health-related outcomes.** To test the association between pSIN and incident health outcomes, all prevalent cases were removed beforehand. The multivariate Cox proportional hazard model provided by the lifeline v.0.27.8 package[62] was used with a pre-set step size of 0.1. Survival outcomes were defined using follow-up time to event and the binary incident event indicator. For all incident outcomes in the whole UKB population, two successive models were tested with an increasing number of covariates: model1 did not adjust for any additional covariate as age and sex had already been regressed out in protein level; model 2 was adjusted for recruitment centre, Townsend deprivation index, IPAQ physical activity group, ethnicity, alcohol frequency, BMI and education years. In current smokers, a third model was applied, which further adjusted for smoking pack years and the number of years smoked. For previous smokers, an additional third model was tested that included adjustments for smoking pack years and the number of years since cessation. *P*-values of the hazard ratio were corrected for FDR multiple testing. Forest plots were generated with a minimum sample size threshold of 80 to ensure adequate statistical power and reliable interpretation.

Cumulative incidence plots were generated utilising the Kaplan-Meier Fitter function from the lifelines package[62]. Due to limitations in case numbers or at-risk numbers at both ends, the x-axis of the plot was constrained to the age range of 45–75. This adjustment ensured a more focused visualisation of the cumulative incidence curve within a clinically relevant age range. *P*-values between cumulative incidence curves were calculated using a log-rank test with adjustment for FDR multiple testing.

A summary of all packages and software used is listed below. CBCgrps [https://cran.r-project.org/web/packages/CBCgrps/index.html], Optuna [https://optuna.org/], SHAP, shap-hypetune [https://github.com/cerlymarco/shap-hypetune], missRanger [https://cran.r-project.org/web/packages/missRanger/index.html], LightGBM [https://lightgbm.readthedocs.io/], scikit-learn [https://scikit-learn.org/], statsmodels [https://doi.org/10.25080/Majora-92bf1922-011], SAIGE, PLINK, SHAPEIT3, IMPUTE2 [https://mathgen.stats.ox.ac.uk/impute/impute_v2.html], FUMA, CTG-VL [https://vl.genoma.io/], clusterProfiler [https://bioconductor.org/packages/release/bioc/html/clusterProfiler.html], lifelines [https://lifelines.readthedocs.io/].

**Reporting summary**
Further information on research design is available in the Nature Portfolio Reporting Summary linked to this article.

## Data availability
The data used in the present study are available from UKB with restrictions applied. Data were used under license and are thus not publicly available. Access to the UKB data can be requested through a standard protocol (https://www.ukbiobank.ac.uk/register-apply/).

The China Kadoorie Biobank (CKB) is a global resource for the investigation of lifestyle, environmental, blood biochemical and genetic factors as determinants of common diseases. The CKB study group is committed to making the cohort data available to the scientific community in China, the UK and worldwide to advance knowledge about the causes, prevention and treatment of disease. To apply, please visit: https://www.ckbiobank.org/data-access. A research proposal will be requested to ensure that any analysis is performed by bona fide researchers. Proteomic data used in this study have restricted access. An anonymised version of the proteomic data analysed in this study is available at https://doi.org/10.6084/m9.figshare.27931350. To get full access to the linked proteomics data, researchers may need to develop formal collaboration with the CKB study group by contacting ckbaccess@ndph.ox.ac.uk. No new data from either UKB or CKB were generated for this study. GWAS summary statistics are provided here (https://g-d7c4c1.dd271.03c0.data.globus.org/pSIN_age_sex_batch_PCs_allChr_ldsc.txt.gz) for open access. Source data are provided with this paper. All other data supporting the findings of this study are available in the article and its Supplementary Information files. Source data are provided with this paper.

## Code availability
All relevant custom code is available for academic use only and can be found at https://github.com/xiaosihao/ProteomicSmoking (a permanent deposit of the code can be found in https://doi.org/10.5281/zenodo.17436647).

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

## Acknowledgements

All UK Biobank data was accessed under UK Biobank Application #61054. This work was supported by the Centre of Artificial Intelligence in Precision Medicines (CAIPM), King Abdulaziz University, Jeddah, Saudi Arabia. We further acknowledge the CKB project staff and the China CDC and its regional offices for assisting with CKB fieldwork. The Genotype-Tissue Expression (GTEx) Project was supported by the Common Fund of the Office of the Director of the National Institutes of Health and by NCI, NHGRI, NHLBI, NIDA, NIMH and NINDS. J.Liu is supported by a University of Oxford Novo Nordisk Research Fellowship. C.L.S. acknowledges support from the NIHR Imperial Biomedical Research Centre (BRC). We further acknowledge the contribution of Yuning Wu for the artwork plotted for the featured image.

## Author contributions

Conceptualization: S.X., B.L., C.v.d., N.A., D.H. Methodology: S.X. Analysis and visualization in UKB: S.X. Analysis and visualization in CKB: S.X., B.L. Analysis and visualization for methylation data: S.W., E.H., J.M. Data management in UKB: S.X., M.A.A., L.B., J.C., J.Liu. Data management in CKB: S.X., B.L., D.B., K.C., L.L., J.Lv, C.Y., D.S., R.C. Supervision: C.v.d., N.A., A.J.N. Writing-original draft: S.X., B.L., N.A., C.v.d. Writing-review & editing: C.v.d., N.A., Z.C., D.H., C.L.S., R.M.M.

## Competing interests

S.X., A.J.N. are funded by the Centre of Artificial Intelligence in Precision Medicines (CAIPM), King Abdulaziz University, Jeddah, Saudi Arabia. C.M.v.D. is supported by the US National Institute on Aging (NIH), NovoNordisk, the Oxford-GSK Institute of Molecular and Computational Medicine (IMCM), Centre of Artificial Intelligence for Precision Medicines (CAIPM) of the University of Oxford and King Abdul Aziz University, Alzheimer Research UK (ARUK), UK National Institute for Health and Care Research (NIHR) Oxford Biomedical Research Center (BRC), ZonMW (Delta Dementie) and Alzheimer Nederland. C.M.v.D. is currently the Research Director, Brain Health of the Health Data Research UK (HDR UK) and the UK Dementia Research Institute (UK DRI), working in partnership with Dementias Platform UK (DPUK). L.L., J.Lv, C.Y., and D.S. are funded by the Chinese National Natural Science Foundation (82192904, 82192903, and 82192900) and the National Key Research and Development Program of China (2016YFC0900500). C.v.d. and N.A. are funded by GSK. The remaining authors declare no competing interests.

## Additional information

¹Nuffield Department of Population Health, University of Oxford, Oxford, UK. ²Centre for Medicines Discovery, Nuffield Department of Medicine, University of Oxford, Oxford, UK. ³Centre of Artificial Intelligence in Precision Medicine (CAIPM), King Abdulaziz University, Jeddah, Saudi Arabia. ⁴Analytic and Translational Genetics Unit, Massachusetts General Hospital, Boston, MA, USA. ⁵Program in Medical and Population Genetics, Broad Institute of MIT and Harvard, Boston, USA. ⁶National Heart and Lung Institute, Imperial College London, London, UK. ⁷National Institute for Health Research (NIHR) Imperial Biomedical Research Centre, London, UK. ⁸Department of Clinical & Biomedical Sciences, University of Exeter Medical School, University of Exeter, Exeter, UK. ⁹Biochemistry Department, Faculty of Science, King Abdulaziz University, Jeddah, Saudi Arabia. ¹⁰Department of Epidemiology & Biostatistics, School of Public Health, Peking University, Beijing, China. ¹¹Peking University Center for Public Health and Epidemic Preparedness and Response, Beijing, China. ¹²Key Laboratory of Epidemiology of Major Diseases, Peking University, Ministry of Education, Beijing, China. ¹³Department of Psychiatry, University of Oxford, Oxford, UK. ✉e-mail: xiao@broadinstitute.org; cornelia.vanduijn@ndph.ox.ac.uk

