## [Transparent Peer Review file · Nature Communications]

Proteomic signatures of smoking and their associations with risk of incident diseases and mortality in diverse populations

Corresponding Author: Dr Sihao Xiao

Version 0:

Reviewer comments:

Reviewer #1

(Remarks to the Author)

Xiao et al developed a protein signature to discriminate smoking status which was predictive of future disease risk in the general population (i.e. including in non-smokers) and could stratify risk among current (and previous) smokers. The analyses are well designed, and the paper is well written. The results are exciting as they pose the possibility to use an objective signature to identify groups at high risk among smokers that would be candidates for more intensive screening and early detection strategies. However, I have some comments particularly regarding some of the more aetiological analyses as it is unclear how these signatures can inform aetiology of smoking, as some of the observed associations may well be driven by independent effects of other factors on the component proteins of the score without any relationship whatsoever to smoking.

Major comments

1. Some more details on the feature selection would be beneficial in the methods section. Specifically, was this done only in the 70% training set?
2. Regarding the generation of the pSIN in the UKB sample. I appreciate the efforts to avoid overfitting through a CV framework. However, assuming feature selection was performed in the training set (70% of current and never smoker as asked in my previous comment?), these people would have already been seen and are therefore the group more prone to overfitting. While the validation in CKB provides confidence in the associations seen, it would be important to clarify this in the methods.
3. More details are needed on the functional enrichment analysis. What did the authors use as a background, i.e. all genes or only those covered by the Olink platform? If using the former, authors should restrict the background to genes covered by Olink as are already over-represented for certain pathways.
4. Lines 145-146: unclear how table s3 related to tissue annotations?? Furthermore, how tissue specific was the expression of these proteins, i.e. albeit they may be expressed in lung, is this expression specific or are these proteins rather expressed throughout many different tissues? Authors can use data from the human protein atlas to look-up specificity of tissue expression.
5. Regarding the GWAS of the pSIN score. To what extent do the authors consider the hits they are getting are related to smoking or related to the proteins in the score but independently of any smoking related mechanisms. The authors report 32 are cis-pQTLs, how many are trans-pQTLs for any of the proteins in the pSIN.
6. Lines 174 – 177: Not sure how figs3 and tables s6-9 relate to colocalization? These just show results from the GWAS but not from colocalization analyses. Table s9 shows some lookups from GWAS catalogue but this does not mean at all that the pSIN and the other traits share the same causal variant. For this formal colocalization analyses need to be performed.
7. Lines 197 – 199: The causal relationships between environmental exposures and smoking are quite complex and some of these associations could be driven by independent effects of the environmental exposures on the proteins that are not due to “smoking related damage”. Authors should discuss this more clearly as a limitation of using such a score to try to understand aetiological mechanisms. This is one of the major limitations of this type of analyses that is very difficult to discern associations that have nothing to do with smoking but are coming up due to independent effects on the proteins composing the pSIN that have nothing to do with smoking.
8. Lines 200 – 216: are these associations (clinical biomarkers and pSIN) similar to the associations between clinical biomarkers and smoking status. Is there any gain in using the pSIN.
9. Regarding the value of pSIN to predict the risk of diseases in current and previous smokers. In my view this is extremely

valuable and exciting as it provides an objective tool to identify a high risk population among smokers (or previous smokers) for early detection and screening. However, it would be important to understand the added values of the pSIN to do so, in comparison to a clinically more acceptable strategy such as using just one biomarker. How does the ability to stratify at risk populations among smokers (or previous smokers) differs when using the pSIN (51 proteins) to when using the single most discriminatory protein of smoking status?

Minor comments:

10. Lines 85 – 89: This may be due to a much lower power and limited set of outcomes considered in these studies rather than a lack of association between the smoking methylation signature and other outcomes (as the sentence suggests)?
11. Line 146 – 147: Quite a lot is known about the association of these proteins and incident diseases actually:
<https://insight.olink.com/data-stories/ukb-diseases>
12. Table s4: the labelling of the traits is unclear, further detail is needed, perhaps a data dictionary.
13. Lines 348 – 350. Duplicated sentences?
14. Unclear which variables were imputed from the description in the methods.
15. Grammar should be revised throughout the methods section to improve readability.

(Remarks on code availability)

Reviewer #2

(Remarks to the Author)

Thank you for the opportunity to review this interesting, comprehensive, and important article. The authors justified the need for their analysis well. They included a large sample for their initial analysis and tested their results using a reasonably large sample in a different demographic population. The figures are clear and help the reader understand the arguments. The authors went through all the expected steps for a high-quality investigation of this nature. My comments are minor in nature. Overall, I was impressed!

1. L143-145: In a sentence or two, clarify how the current project leveraged the GTEx project. Methodologically, how were the findings of Lonsdale et al. used to create Figure s2?
2. Table S4: Clarify whether all variables were analyzed together in a single regression model. It would not be appropriate to include all of smoking years, pack years, and number of cigarettes together in one model. Similarly, it would not be appropriate to include smoking cessation years as a continuous variable and as a categorical variable together in one model.
3. Table S5/L166-168: Comment that the trends between CKB and UKB are more consistent for current smokers than for former smokers.
4. L169-171: Very briefly, provide methodological details about how the GWAS was conducted such that the reader does not need to go to the end of the article to the Methods. What was the outcome being associated? Also clarify if this was done in UKB only.
5. L172-L174: Add citations for the referenced studies.
6. L187-188: Very briefly, provide methodological details about how the exposome analysis was conducted. Were all factors considered together in a single model? What statistical approach was taken? (I realize details are in the Methods, but help guide the readers through the many logical steps you took – it's excellent, but a lot!).
7. L103 and L240: Inconsistent number of major disease outcomes reported.
8. L242: Add the second decimal place to the upper confidence interval limit.
9. L308-311: Can be reworded for clarity.
10. Figure S10 and s11: Clarify the reference group. Is this saying that higher pSIN scores are protective among these groups? Or is this in comparison to people with high-risk previous smokers (s10) and high-risk current smokers (s11)?
11. In the discussion, comment on the fact that the external validation study was limited to only males, and that the health association analysis was not validated in this population.
12. L461: Comment on the % missingness for key variables.
13. L503: Justify why "occasional smokers" were considered never smokers.
14. For the regression models throughout the paper, justify how covariates were chosen (e.g., section starting L583).

(Remarks on code availability)

NA

Reviewer #3

(Remarks to the Author)

This is an exemplary piece of analytical research into Proteomic Olink signatures and their association with multiple diseases.

The raw data was taken from the UKBB and the validation undertaken with the China Kadoorie Biobank, The AUC values are indeed very high, far higher than many similar biomarker analysis.

The sub-analysis was not just with smokers and never smokers but also with environmental exposures, health status (ie. obesity), sociodemographic and lifestyle factors and mortality data. The pSIN parameter was also used to differentiate

recovery status of previous smokers, which is a completely new and impressive approach.

The indepth details provided for machine learning analysis and statistical tools are well described.

This is an academic exercise providing a very informative manuscript. The authors have also included a number of possible limitations in their discussion, even indicating that the study has only limited statistical power to detect associations between pSIN and common diseases.

However, even considering all the positive elements provided in this analysis, there is no attempt to indicate how it maybe used or implemented in clinical practice. We are now in an era of not just discovery research but also translational research. The authors should have pencilled in a possible way their findings could be tested in a primary care clinical trial and the costs involved in utilising the 51 proteomic biomarkers (Olink or another platform) and the collection of the wide range of clinical, epidemiological, environmental data. And if this is a realistic way forward for identification of high-risk individuals?

(Remarks on code availability)

Reviewer #4

(Remarks to the Author)

The work provides the first affinity proteomics signature to differentiate between “never smoking” and “currently smoking” subgroups. The signature is based on a machine learning model trained with gradient boosting and subsequent feature selection, resulting in a signature of 51 selected proteins. Based on this gradient boosting trained machine learning model, a smoking Index (pSIN) is defined. It is encouraging to see that within the 51 proteins around 5-10 are highly expressed in the lung, and other relevant tissues, while the signature is obtained from plasma samples. The proteins and pathways associated with the signature give a novel perspective on underlying biological changes, induced by smoking. Note that true causality is or course difficult to show given the study set up. Nevertheless, there is external validation for the male population based on an external cohort. The test also provides a novel and different perspective on self-reporting, and methylation profiles for smoking. Where the previously reported methylation profiles seem to be more closely associated with smoking history, while the plasma profiles seem to reflect the response to smoking – as the reported index becomes continuously lower after smoking cessation. Overall, the study provides a novel perspective on the bodies smoking response, and has the potential to monitor the impact of smoking on the body.

Major issues:

1. Firstly, the added value of the proteomics signature, over the self-reported smoking and the existing methylation signature, should be made more clear. In addition, the added value should be highlighted more where contrastive results have already been provided (for example, the difference between methylation and proteomics profile in the “previous smoker group”). Moreover, in all associations reported with the pSIN value, the strength of associations with the self-reported smoking status, and methylation profile should be provided as a reference, to make the novelty and added value of the proteomics signature more clear; this includes the association analysis of major diseases and mortalities, genomics factors /GWAS results, haematological measures, social-demographic and lifestyle factors.
2. The labels used to train the machine learning model (“current smokers” vs “never-smoker”) are extremely imbalanced in the dataset. This can give biases in several reporting measure of classification measures, including the AUC-ROC. A wider range of performance measures should be reported (both on validation and test set), including: F1, balanced accuracy, and the AUC-PR/RC. In addition, one can undersample the majority class in the held-out test set, to get a better performance estimate.
3. The argument that the signature cannot be validated on the 2137 female smoking participants does not hold. It is possible to report accuracies and AUC ROC statistics on as little as 50–100 samples. Presumably, the signature does not translate to the female cohort. Instead of hiding this, the authors should report the separate accuracy statistics for the female only test external validation set, as well as a separate performance analysis on the female subset of the UK biobank cohort. If there are differences, and the signature does not translate well to female subjects, this is in itself interesting, and should be reported fairly.
4. Missing data was imputed. Please describe more clearly what data was imputed, and in which analysis imputed data was used. It is essential that in the test data set, over which the performance of the classification model was measured, no imputed data was used for the labels (i.e. current smokers vs non-smokers). In addition, the variables tested for association with pSIN should be free from imputation or at least it should be tested if the association holds when imputation is not performed.

Minor issues:

1. In the abstract, the added value of pSIN with respect to self reported smoking history, and the methylation profile should be made more clear.
2. In the abstract, the rationale behind the genetic variant associations is completely lost. (Why would one expect a link

between a lifestyle choice of smoking and genetics?). Perhaps take this out if there is no sufficient space to explain this. It is very confusing to report this without appropriate context.

3. Objectivity of signature:

a. Page 3, line 71/72 "... there is an urgent need to develop objective measures ..."

b. page 13 line 272 "This suggests pSIN can serve as an objective test ..."

Making claims about objectivity, should be accompanied by a discussion on the accuracy of the model. What if this signature is used to check someone's smoking history for insurance purposes? What guarantee can be provided that someone with a high pSIN score has indeed smoked? Make the limitations of the model more clear, when objectivity of the model is claimed.

4. "age" and "sex" are regressed out of the proteome expression data. As this is done via a linear model, and the machine learning model (gradient boosting) is non-linear, one cannot assume that there is no remaining signal left in the data to predict age or sex on the regression data. One way to test if the signal is taken out completely, is by training a classifier to predict sex or age on the regressed data set. Alternatively, one can discuss the impact on the results and signature if the regression does not remove all associations with age and sex in the proteome expression data.

5. Please explain how the model probability score for a given class that comes out of the classification model is transformed to a pSIN score with a range of approximately -10.0-10.0. Make in the text of the results section more clear what a pSIN score of 0.0 indicates, or more generally how the reader can quantitatively interpret the pSIN score.

6. Very strong associations between pSIN and haematological measurements and the blood biochemical composition are found (S4 and S5). Is there any clear link between the proteins found in the plasma based pSIN signature, and these findings? Would any of these measures by itself be sufficient to classify "current smokers" and "never-smokers".

7. P6 line 178-183. It is unclear what the rationale is for performing these genetic correlations. Are you looking for risk factors to start smoking? Or are you interested to see which genetic variants can worsen the effects of smoking, or perhaps cause a smoking-like signature in non-smokers? Make the aim of the analysis more clear, and justify the choice of datasets to ask this question.

8. It may be helpful to the reader to explain better how the "smoking history" has been collected. Is this self-reported? P8 line 226: explain more clearly why it makes sense to look at variance explained from "smoking history" in a profile that was learned to classify current smokers vs never-smokers.

9. P22 "Feature interpretation and selection" it should be clearly described over which data the feature selection and hyperparameter tuning was performed. Figure 1 suggests that this is done over the train/validation set and not over the hold-out test set; as it should be. However, this should be described explicitly in the text as well.

(Remarks on code availability)

The published code is not sufficient to redo the analysis. The readme file is not sufficient. It is not clear which input files are required to run the script. The code is poorly annotated / commented. Internal code review is required, before resubmission.

Version 2:

Reviewer comments:

Reviewer #1

(Remarks to the Author)

The authors have addressed all comments nicely. I only have one outstanding comment relating to my previous comment #8. Authors have performed additional sensitivity analyses adjusting associations between biomarkers or risk factor and pSIN by smoking status, and conclude : "while pSIN captures smoking-related biological signatures, it also provides additional information beyond conventional smoking status, particularly for certain biomarkers and health outcomes". It would be important to understand to what extent pSIN is capturing other biological factors if there were to be considered for any downstream translational application. Therefore, I strongly suggest authors perform a more in-depth analysis looking at the proportion of pSIN variance explained by a comprehensive set of biological (prevalent disease status, clinical biomarkers, anthropometric, lifestyle (including smoking status), dietary and demographic factors), genetic (e.g. PRS for smoking, COPD, lung cancer, etc.) and technical variables (potentially affecting proteome measurements, e.g. fasting time, study centre, etc.), when all variables compete in the same model.

(Remarks on code availability)

Reviewer #4

(Remarks to the Author)

The clarity of the manuscript has much improved due to the revisions. In particular, the value and impact of the pSIN profile are much more explicitly described, and the added value is more clearly demonstrated by the additional analyses.

One minor issue remains:

Comment Reviewer 1, comment 3:

The background of the GO enrichment analysis may still be problematic. Olink panels are not a set of randomly chosen proteins (unlike RNAseq or untargeted MS proteomics), but are a selected set of proteins based on disease profiles. ClusterProfiler assumes an untargeted omics set. It would be good, to use a method, that can take the Olink target panel as background into account, otherwise pathways may come up, that were simply selected within the Olink panels.

(Remarks on code availability)

Some parts of the code are still quite poorly annotated, but the overall structure is now clear.

Response to reviewers' comments (Manuscript no.: 551097)

We would like to thank external reviewers for their detailed consideration and helpful feedback which has helped improve the manuscript. We have incorporated the reviewers' comments and expanded our methodology, results, and discussion to reflect these requests. Responses to the reviewers' comments (in bold) are detailed below, along with the accompanying revisions to the paper.

Reviewer #1:

Xiao et al developed a protein signature to discriminate smoking status which was predictive of future disease risk in the general population (i.e. including non-smokers) and could stratify risk among current (and previous) smokers. The analyses are well-designed, and the paper is well-written. The results are exciting as they pose the possibility to use an objective signature to identify groups at high risk among smokers that would be candidates for more intensive screening and early detection strategies. However, I have some comments particularly regarding some of the more aetiological analyses as it is unclear how these signatures can inform aetiology of smoking, as some of the observed associations may well be driven by independent effects of other factors on the component proteins of the score without any relationship whatsoever to smoking.

We thank the reviewer for their time and positive feedback. We have incorporated the suggestions of the reviewer in the revised version of the manuscript.

Major comments

1. Some more details on the feature selection would be beneficial in the methods section. Specifically, was this done only in the 70% training set?

Response: We apologize for the omission of details. We have added the following details to the Methods section in the revised version of the manuscript (lines 634-647):
“The Boruta algorithm enhances feature selection by creating randomly permuted shadow features that serve as a baseline for comparison. Specifically, shadow features are duplicates of the original features with their values randomly shuffled to break any associations with the target variable. The algorithm compares the mean absolute SHAP values of the original features against those of the shadow features. Features are retained only if they demonstrate a higher mean SHAP value than the highest-scoring

shadow feature, thereby ensuring their predictive importance is not due to random chance. In our study, the algorithm was executed for 200 iterations to ensure robustness and stability in feature ranking. Features falling within the bottom 5% of importance scores (tail 5%) were systematically rejected as they were unlikely to contribute meaningfully to the model's performance. Following the feature selection process, the refined model—consisting of Boruta-selected features—underwent another round of hyperparameter tuning to optimize its performance before further analysis. All model tuning and feature selection steps were performed within the 70% training set in UKB.”.

2. Regarding the generation of the pSIN in the UKB sample. I appreciate the efforts to avoid overfitting through a CV framework. However, assuming feature selection was performed in the training set (70% of current and never smokers as asked in my previous comment?), these people would have already been seen and are therefore the group more prone to overfitting. While the validation in CKB provides confidence in the associations seen, it would be important to clarify this in the methods.

Response: We agree that external validation is a more robust way of testing overfitting. As suggested, we have now provided further clarification in the Methods section lines 671-677. *“This approach allowed for a more robust estimation of the impact of smoking on protein profiles across the UK Biobank cohort compared to using only one model trained on training data to calculate pSIN for the entire population. External validation in CKB was performed to further test the possibility of the overfitting problem. Although the hyperparameter tuning step still gave the possibility of data leakage and potentially overfitting problems, we further performed external validation. For external validation, a model with the optimized hyperparameter was trained in the UKB training dataset and was tested in the CKB. Performances of identifying current smokers from never smokers were compared.”*

3. More details are needed on the functional enrichment analysis. What did the authors use as a background, i.e. all genes or only those covered by the Olink platform? If using the former, authors should restrict the background to genes covered by Olink as are already over-represented for certain pathways.

Response: For clarification, the pathway annotations in Lines 152–155 were derived from GO molecular functions assigned to each protein using the *clusterProfiler* package. However, no formal enrichment analysis was performed. Instead, we focused on annotating molecular functions for individual proteins rather than assessing pathway overrepresentation relative to a background set. We have updated the Methods section lines 681-689 and changed the subtitle into function annotation for proteins to make this clearer.

4. Lines 145-146: unclear how table s3 related to tissue annotations??

Furthermore, how tissue specific was the expression of these proteins, i.e. albeit they may be expressed in lung, is this expression specific or are these proteins rather expressed throughout many different tissues? Authors can use data from the human protein atlas to look-up specificity of tissue expression.

Response: We apologize for the confusion. Table s3 was meant to show the annotations of the Boruta-selected proteins which should be referenced earlier in the text. Tissue enrichment analysis was performed based on the GTEX tissue mRNA expression levels using the FUMA web tool (see method). Two plots were provided to show the mRNA expression levels of each gene of the selected protein in available tissues and in which tissue does genes of selected proteins are differentially expressed. Protein expression data available on human protein atlas does not provide quantitative measurements, instead only four category values (i.e., low, medium, high and not detected) are provided. The human protein atlas website has also been using tissue RNA expression from GTEX as a reference for tissue specificity analysis in the protein summary section.

5. Regarding the GWAS of the pSIN score. To what extent do the authors consider the hits they are getting are related to smoking or related to the proteins in the score but independently of any smoking related mechanisms. The authors report 32 are cis-pQTLs, how many are trans-pQTLs for any of the proteins in the pSIN.

Response: Of the 129 genes identified to be associated with pSIN, 75 were novel as in they were not associated with smoking initiation before in both genetic and epigenetic association studies. LD score regression analysis shows the strongest correlation with current smoking ($r^2 = 0.78$) followed by lung cancer ($r^2=0.71$) and COPD ($r^2=0.54$), which suggests that pSIN GWAS is mainly capturing smoking and/or related mechanisms.

For the second question, we have now added the following details in the result section lines 190-196: *Of the 95 significant lead independent variants, 34 were cis-pQTLs mapping to 16 genes, 8 of which encoded proteins included in the pSIN model. Six cis-pQTLs were annotated for multiple closely located genes simultaneously (e.g., rs901886 and rs35929247 were cis-pQTLs for ICAM1, ICAM3, ICAM4, and ICAM5). Additionally, 7 variants were identified as trans-pQTLs mapping to 366 genes, 21 of which encoded proteins selected in the pSIN model. Among these, rs2519093 showed the broadest mapping, acting as a trans-pQTL for 358 genes and a cis-pQTL for two genes (ABO and DBH).*

6. Lines 174 – 177: Not sure how figs3 and tables s6-9 relate to colocalization?

These just show results from the GWAS but not from colocalization analyses. Table s9 shows some lookups from GWAS catalogue but this does not mean at all that the pSIN and the other traits share the same causal variant. For this formal colocalization analyses need to be performed.

Response: We apologize for incorrectly using the word “colocalization”, we did not perform formal colocalization analyses, as our focus was on identifying associations rather than establishing causal relationships between these variants and traits. To ensure accuracy, we have revised the text to avoid the term "colocalization" which now states (line196-201):

" Of the 129 genes associated with pSIN, 10 genes were previously identified as GWAS smoking loci^{9,21}, and 54 genes were previously found in epigenetic studies^{10,14,22}, 75 (58%) were novel and have previously been reported to be associated with body mass index (BMI), diabetes, cancer development, and immunological/haematological traits including lymphocyte counts, eosinophil counts, and white blood cell counts (Fig S3, Table s6-s9)."

7. Lines 197 – 199: The causal relationships between environmental exposures and smoking are quite complex and some of these associations could be driven by independent effects of the environmental exposures on the proteins that are not due to “smoking related damage”. Authors should discuss this more clearly as a limitation of using such a score to try to understand aetiological mechanisms. This is one of the major limitations of this type of analysis that is very difficult to discern associations that have nothing to do with smoking but are coming up due to independent effects on the proteins composing the pSIN that have nothing to do with smoking.

Response: Thanks for your comments. We did not declare a causal relationship in the manuscript. However, to make this clearer, we have now expanded the discussion on study limitation and added the following (lines 457-467): *“The causal relationships between environmental exposures and smoking are inherently complex. While our analysis aimed to leverage protein signatures within pSIN to understand the downstream pathology of smoking, it is essential to acknowledge that some associations identified may not be directly attributable to smoking behaviour. Instead, these associations could arise from independent effects of environmental exposures and smoking-related behaviours on proteins that are part of the pSIN. This limitation underscores the challenges of disentangling causal pathways in observational data and highlights the necessity for cautious interpretation when utilizing scores based on protein networks to elucidate aetiological mechanisms. Future studies should consider complementary approaches, such as Mendelian randomization or experimental validation, to disentangle these independent effects and strengthen causal inferences.”*

8. Lines 200 – 216: are this associations (clinical biomarkers and pSIN) similar to

the associations between clinical biomarkers and smoking status. Is there any gain in using the pSIN.

Response: To answer this question, we conducted additional sensitivity analyses where the associations between pSIN and clinical biomarkers were further adjusted for smoking status (Figure below) and added to the result session (lines 243-250) (Fig S6a,b). We observed that while the effect sizes of biomarkers such as HbA1c, GlycA, and Triglycerides were attenuated, they remained statistically significant. However, biomarkers such as APOA, HDL cholesterol, and Creatinine became non-significant after this adjustment. Similarly, the association between pSIN and clinical risk factors, such as poor self-rated health, was weakened when adjusting for smoking status while the associations with other risk factors remained unchanged. These findings suggest that while pSIN captures smoking-related biological signatures, it also provides additional information beyond conventional smoking status, particularly for certain biomarkers and health outcomes. We have now further expanded the results and discussion sections on the findings of additional sensitivity analyses.

Association between clinical biomarkers and risk factors to pSIN

9. Regarding the value of pSIN to predict the risk of diseases in current and previous smokers. In my view this is extremely valuable and exciting as it

provides an objective tool to identify a high risk population among smokers (or previous smokers) for early detection and screening. However, it would be important to understand the added values of the pSIN to do so, in comparison to a clinically more acceptable strategy such as using just one biomarker. How does the ability to stratify at risk populations among smokers (or previous smokers) differs when using the pSIN (51 proteins) to when using the single most discriminatory protein of smoking status?

Response: Thank you for your insightful comments. To assess the added value of pSIN compared to using a single biomarker, we have evaluated the predictive performance of the top three proteins ranked by SHAP value within the UKB test dataset with pSIN

(Figure below). ALPP and CXCL17 demonstrated relatively strong discrimination with AUCs of 0.88 and 0.87, respectively, while ACVRL1 showed lower performance with an AUC of 0.76. However, none of these individual proteins matched the predictive power of pSIN, which achieved an AUC of 0.95. These results highlight the advantage of using a multi-protein composite score over a single biomarker for more accurate stratification of current smokers and never smokers. We have added this to the result session (lines 132-139) (Fig s1c).

Minor comments:

10. Lines 85 – 89: This may be due to a much lower power and limited set of outcomes considered in these studies rather than a lack of association between the smoking methylation signature and other outcomes (as the sentence suggests)?

Response: This could be a possible explanation as most studies of DNA methylation scores on smoking only consisted of several thousand participants which is magnitude smaller than our study. It could also be that most of these cohorts also have short

follow-up times which result in short incident disease cases as well. This has been further discussed in the discussion section lines 397-400.

11. Line 146 – 147: Quite a lot is known about the association of these proteins and incident diseases actually: <https://insight.olink.com/data-stories/ukb-diseases>

Response: We would like to thank the reviewer for pointing out this resource. We have now changed to: ALPP is synthesized in the liver by a metalloenzyme that catalyzes the hydrolysis of phosphoric acid monoesters and was previously found to be associated with COPD and cancers. (lines 158-160)

12. Table s4: the labelling of the traits is unclear, further detail is needed, perhaps a data dictionary.

Response: As suggested, we have now revised the table with better labelling.

13. Lines 348 – 350. Duplicated sentences?

Response: We have removed the duplicated sentence.

14. Unclear which variables were imputed from the description in the methods.

Response: For clarification, we did not impute any exposure or outcome variables but only co-variables of the linear association between biomarkers and pSIN, clinical risk factors and pSIN, GWAS, exposome analysis and multivariable Cox model. We have changed the relevant sentence to: “*Missing data was imputed using a combined method with random-forest-based algorithm and predictive mean matching provided by R package missRanger¹ when used as a covariate in the linear association between biomarkers and pSIN, clinical risk factors and pSIN, GWAS, exposome analysis and multivariable cox model (Townsend deprivation index, IPAQ physical activity group, ethnicity, alcohol frequency, BMI, and education years)*”. (line 534-540)

15. Grammar should be revised throughout the methods section to improve readability.

Response: The grammar and clarity of the methods have been revised.

Reviewer #2:

Thank you for the opportunity to review this interesting, comprehensive, and important article. The authors justified the need for their analysis well. They

included a large sample for their initial analysis and tested their results using a reasonably large sample in a different demographic population. The figures are clear and help the reader understand the arguments. The authors went through all the expected steps for a high-quality investigation of this nature. My comments are minor in nature. Overall, I was impressed!

We would like to thank the reviewer for their time, input and positive feedback. We have made changes to the manuscript according to their suggestions.

1. L143-145: In a sentence or two, clarify how the current project leveraged the GTEx project. Methodologically, how were the findings of Lonsdale et al. used to create Figure s2?

Response: For improved clarity, we have now revised the relevant sentence to: *“Tissue enrichment analysis was performed among selected proteins using RNA expression data from Genotype-Tissue Expression (GTEx) project². Results indicated that many of these proteins were differentially expressed in tissues either directly exposed to or affected by smoking including lung, salivary glands, colon, esophagus and adipose tissues.” In line 152-155*

Figure s2 was created using the method stated in the Functional enrichment analysis of the method section *line(677-683): Tissue-specific protein expression data was extracted from the Genotype-Tissue Expression (GTEx) project² database v.8 and the heatmap was plotted using FUMAGWAS webtool. Values showed on the heatmap were the average of normalized expression per gene. Differential expression genes (DEG) were identified by a 2-sided t-test per tissue type versus all other tissue types. Genes with a Bonferroni corrected p-value < 0.05 and absolute log fold change > 0.58 were selected as DEG and were shown as red colour. This is also added to Fig s2 description.*

2. Table S4: Clarify whether all variables were analyzed together in a single regression model. It would not be appropriate to include all of smoking years, pack years, and number of cigarettes together in one model. Similarly, it would not be appropriate to include smoking cessation years as a continuous variable and as a categorical variable together in one model.

Response: We apologize for the lack of clarity. Each smoking-related exposure was considered separately. Similarly, smoking cessation years were analyzed separately as either continuous or categorical variables in distinct models, with categorical analyses designed to explore dose-response relationships. The supplementary table was redesigned to make this clearer. In the revised version we have made it clearer (lines 694-713)

3. Table S5/L166-168: Comment that the trends between CKB and UKB are more consistent for current smokers than for former smokers.

Response: We have added the following text to the results section lines 175-183: *“Similar trends were observed in CKB current smokers, where smoking duration, smoking exposure, and smoking intensity were all significantly and positively associated with pSIN. However, among former smokers, significant associations were observed only for years since smoking cessation and smoking duration, whereas associations with smoking intensity and smoking exposure were not statistically significant. This may be because there are more modifiers related to the difference in behavioural or epidemiological characteristics between Chinese and UK populations such as relapse after quitting smoking in previous smokers than in current smokers.”*

4. L169-171: Very briefly, provide methodological details about how the GWAS was conducted such that the reader does not need to go to the end of the article to the Methods. What was the outcome being associated? Also clarify if this was done in UKB only.

Response: We have added the following text to the results lines 184-187: *“To understand the genetic architecture underlying the biological consequences of smoking pathology, as indexed by pSIN, we performed a genome-wide association study (GWAS) of pSIN in the UKB. We applied a linear mixed model in SAIGE, adjusting for age, sex, genotyping batch effects, and the first 40 principal components to account for population structure.”*

5. L172-L174: Add citations for the referenced studies.

Response: We apologize for the omission. We have added the references in the revised version of the manuscript.

6. L187-188: Very briefly, provide methodological details about how the exposome analysis was conducted. Were all factors considered together in a single model? What statistical approach was taken? (I realize details are in the Methods, but help guide the readers through the many logical steps you took – it’s excellent, but a lot!).

Response: We have added the following text to the revised version of the manuscript: *“The exposome-wide analysis was conducted using generalized linear models, where each factor was tested separately for its association with pSIN. Models were adjusted for recruitment centre, ethnicity, and smoking status.”*. Lines 212-215.

7. L103 and L240: Inconsistent number of major disease outcomes reported.

Response: For confirmation, the number in the L240 is correct. We have now changed L103 into: “*Finally, we examined the associations between our proteomic-based smoking score and clinical risk factors, blood-based biomarkers, and risks of 27 major diseases and mortality in the UK Biobank*”.

8. L242: Add the second decimal place to the upper confidence interval limit.

Response: We have added the second decimal point for lung cancer (HR=1.97, CI:1.83, 2.11) (line 276)

9. L308-311: Can be reworded for clarity.

Response: We have rephrased the entire paragraph lines 340-346.

10. Figure S10 and s11: Clarify the reference group. Is this saying that higher pSIN scores are protective among these groups? Or is this in comparison to people with high-risk previous smokers (s10) and high-risk current smokers (s11)?

Response: As requested, we have now clarified the reference groups in Figures S10 and S11. In Figure S10, the comparisons are made between previous smokers with low pSIN (i.e., those whose proteomic profiles resemble never-smokers) and high-risk previous smokers with elevated pSIN scores. In Figure S11, the comparisons are between current smokers with low pSIN (whose proteomic profiles are similar to never smokers) and high-risk current smokers with high pSIN.

The forest plot shows that those with proteomic profiles similar to never smokers indeed had lower incident disease risks. The observed differences in health outcomes highlight the ability of pSIN to stratify risk beyond traditional smoking classifications.

We have updated the figure legends to ensure this distinction is clear.

11. In the discussion, comment on the fact that the external validation study was limited to only males, and that the health association analysis was not validated in this population.

Response: Thank you for your comment. Initially, we limited the external validation in CKB to males due to the very low prevalence of smoking among females in this cohort. However, we recognize that restricting validation to only one sex may limit the generalizability of our findings. To address this, we have now expanded the validation to include both male and female participants. The model trained in the UKB training dataset was re-evaluated in the full CKB cohort, achieving an AUC of 0.91. Additionally, major diseases with a sufficient number of cases (including lung cancer, COPD, stroke, respiratory disease (combined), and mortality) remained significantly associated with

pSIN in Cox proportional hazards models, further supporting the robustness of our findings.

We have updated the result and discussion to match the new results Lines (100, 114-116, 139-142, 144-146, 284-287)

12. L461: Comment on the % missingness for key variables.

Response: We have added the % missingness of key variables to the methods section lines 534-540.

13. L503: Justify why “occasional smokers” were considered never smokers.

Response: We classified "occasional smokers" as never smokers to ensure consistency in our comparison groups and to align with conventions in CKB, UKB and prior epidemiological studies³⁻⁶, in which only those who had smoked at least one cigarette a day persistently for at least 6 months were considered as smokers. Occasional smokers comprised a very small proportion of the population and did not report a sustained smoking history, with similar levels of exhaled CO levels in CKB to that among never smokers. Given that our study focuses on the long-term molecular impact of regular smoking, we categorized occasional smokers with never smokers to avoid potential misclassification bias and to maintain a clear distinction between individuals with significant smoking exposure and those without.

We have now added this justification in the text lines 514-520 to clarify our approach.

14. For the regression models throughout the paper, justify how covariates were chosen (e.g., section starting L583).

Response: In our analyses, the selection of covariates for the regression models was based on a combination of prior literature, biological plausibility, and statistical considerations to minimize confounding while avoiding overadjustment. To make the methodology clearer, this has been changed to (lines 694-713):

For associations between smoking history and pSIN, models were adjusted for basic socioeconomic factors including recruitment centre, ethnicity, education years, and Townsend deprivation index. Each smoking-related exposure was considered separately. Similarly, smoking cessation years were analyzed separately as either continuous or categorical variables in distinct models, with categorical analyses designed to explore dose-response relationships.

For associations between blood biomarkers/clinical risk factors/haematological measurements to pSIN, continuous exposure variables, standardization was applied before inclusion in the models. Associations were adjusted additionally for lifestyle factors including IPAQ activity group and alcohol intake frequency as they are known to confound physiology status and liver damage. To explore the added value of pSIN

compared to self-reported smoking status, sensitivity analysis was performed adjusting additionally for self-reported smoking status.

For exposome-wide association analysis (all available social-economic and lifestyle variables available in UKB), models were only adjusted for the most basic recruitment centre, ethnicity and smoking status to allow exploration of a wide range of potential associations between the exposome and pSIN without masking potential signals by controlling for too many variables.

Social-economic and lifestyle confounding factors were chosen based on previous literatures⁵⁴⁻⁵⁶.

Reviewer #2 (Remarks on code availability):

NA

Reviewer #3 (Remarks to the Author):

This is an exemplary piece of analytical research into Proteomic Olink signatures and their association with multiple diseases.

The raw data was taken from the UKBB and the validation undertaken with the China Kadoorie Biobank, The AUC values are indeed very high, far higher than many similar biomarker analysis.

The sub-analysis was not just with smokers and never smokers but also with environmental exposures, health status (ie. obesity), sociodemographic and lifestyle factors and mortality data. The pSIN parameter was also used to differentiate recovery status of previous smokers, which is a completely new and impressive approach.

The indepth details provided for machine learning analysis and statistical tools are well described.

This is an academic exercise providing a very informative manuscript. The authors have also included a number of possible limitations in their discussion, even indicating that the study has only limited statistical power to detect associations between pSIN and common diseases.

However, even considering all the positive elements provided in this analysis, there is no attempt to indicate how it maybe used or implemented in clinical practice. We are now in an era of not just discovery research but also translational research. The authors should have pencilled in a possible way their

findings could be tested in a primary care clinical trial and the costs involved in utilising the 51 proteomic biomarkers (Olink or another platform) and the collection of the wide range of clinical, epidemiological, environmental data. And this is a realistic way forward for identification of high-risk individuals?

Response: We thank the reviewer for their time, input and positive feedback. We acknowledge that while our study provides a strong foundation for understanding the molecular signatures of smoking and its associations with disease risk, further steps are needed to explore its potential clinical applications.

To bridge the gap between discovery and translational research, we propose that future studies should evaluate the feasibility of integrating pSIN into primary care settings. This could involve testing its utility in a clinical trial designed to assess whether proteomic-based risk stratification improves early detection and targeted interventions for high-risk individuals. Key considerations include the cost-effectiveness of measuring the 51 proteomic biomarkers using Olink or alternative platforms, as well as the logistical challenges of incorporating these assessments into routine clinical workflows.

Additionally, implementing pSIN in practice would require careful evaluation of whether it provides actionable information beyond traditional risk factors, and whether its use could justify the costs associated with proteomic profiling and the collection of comprehensive clinical, epidemiological, and environmental data. While high-throughput proteomics is currently expensive, costs are expected to decline with technological advancements, potentially making it a viable tool for personalized risk assessment in the future.

We have now included a discussion of these considerations in the manuscript to highlight possible pathways for translation into clinical practice. Lines (468-486)

Reviewer #4 (Remarks to the Author):

The work provides the first affinity proteomics signature to differentiate between “never smoking” and “currently smoking” subgroups. The signature is based on a machine learning model trained with gradient boosting and subsequent feature selection, resulting in a signature of 51 selected proteins. Based on this gradient boosting trained machine learning model, a smoking Index (pSIN) is defined. It is encouraging to see that within the 51 proteins around 5-10 are highly expressed in the lung, and other relevant tissues, while the signature is obtained from plasma samples. The proteins and pathways associated with the signature give a novel perspective on underlying biological changes, induced by smoking. Note that true causality is of course difficult to show given the study set up.

Nevertheless, there is external validation for the male population based on an external cohort. The test also provides a novel and different perspective on self-reporting, and methylation profiles for smoking. Where the previously reported methylation profiles seem to be more closely associated with smoking history, while the plasma profiles seem to reflect the response to smoking – as the reported index becomes continuously lower after smoking cessation. Overall, the

study provides a novel perspective on the bodies smoking response, and has the potential to monitor the impact of smoking on the body.

Major issues:

1. Firstly, the added value of the proteomics signature, over the self-reported smoking and the existing methylation signature, should be made more clear. In addition, the added value should be highlighted more where contrastive results have already been provided (for example, the difference between methylation and proteomics profile in the “previous smoker group”). Moreover, in all associations reported with the pSIN value, the strength of associations with the self-reported smoking status, and methylation profile should be provided as a reference, to make the novelty and added value of the proteomics signature more clear; this includes the association analysis of major diseases and mortalities, genomics factors /GWAS results, haematological measures, social-demographic and lifestyle factors.

Response: We would like to thank the reviewer for their time and helpful comments. We agree that the added value of the pSIN over self-reported smoking status and the existing methylation signature should be made clearer, particularly in association analyses. To address this, we have performed additional sensitivity analyses, adjusting for smoking status in associations with haematological measures, biological markers, clinical risk factors, and social-demographic and lifestyle factors. These analyses help to better isolate the specific contribution of pSIN, independent of self-reported smoking status. (line 243-250) (Fig s6a,b)

Regarding associations with incident disease outcomes, we have already conducted analyses stratified by smoking status, with further adjustments for variables such as smoking duration, smoking exposure, and smoking cessation. This approach serves a similar purpose, ensuring that the associations we observe with pSIN do not simply reflect the effects of smoking history.

In the GWAS analysis, we focused on identifying all genomic factors associated with pSIN, including those linked to smoking histories, smoking-related diseases, and other lifestyle factors. While we did not adjust for smoking status in this context, we did highlight that 58% of the significantly associated variants were not found to be linked to smoking initiation in previous genetic or epigenetic studies. This reinforces the novelty of the pSIN signature, showing that a substantial portion of the genetic variants associated with pSIN are independent of smoking initiation, and may reflect broader biological processes.

For comparison with methylation, unfortunately, there were currently no epigenetic data available in either the UKB or the CKB. Additionally, the existing epigenetic cohorts did not have proteomic data, and usually, these cohorts tend to have smaller sample sizes and shorter follow-up periods, making it challenging to establish the link between epigenetic signatures of smoking and future risks of smoking-related diseases. However, this limitation also presents one of the key advantages of our research, as the combination of proteomics with a large sample size and long follow-up period in UKB

and CKB offers a more comprehensive approach to understanding the biological consequences of smoking.

We have updated the manuscript to include these clarifications, ensuring that the added value of the proteomics signature is highlighted in comparison to self-reported smoking status and methylation profiles and that the results from the additional analyses are presented.

2. The labels used to train the machine learning model (“current smokers” vs “never-smoker”) are extremely imbalanced in the dataset. This can give biases in several reporting measure of classification measures, including the AUC-ROC. A wider range of performance measures should be reported (both on validation and test set), including: F1, balanced accuracy, and the AUC-PR/RC. In addition, one can undersample the majority class in the held-out test set, to get a better performance estimate.

Response: To address the issues raised, we have now added F1, balanced accuracy and the averaged precision score in addition to AUC to give a more comprehensive assessment of the model performance. We have added all scoring matrices in the result section. Lines 122-131, lines 140-141.

3. The argument that the signature cannot be validated on the 2137 female smoking participants does not hold. It is possible to report accuracies and AUC ROC statistics on as little as 50–100 samples. Presumably, the signature does not translate to the female cohort. Instead of hiding this, the authors should report the separate accuracy statistics for the female only test external validation set, as well as a separate performance analysis on the female subset of the UK biobank cohort. If there are differences, and the signature does not translate well to female subjects, this is in itself interesting, and should be reported fairly.

Response: Please see our response to Reviewer 2’s 11th comments on the same issue. We have now re-validated the model using the entire CKB cohort, which includes both male and female participants. The UKB model derived in the training set was subsequently externally validated in CKB, achieving an AUC of 0.91 (SD=2.22*10⁻¹⁶; **Fig 2a**) (F1=0.93, SD=1.11*10⁻¹⁶; AP=0.87, SD=1.11*10⁻¹⁶; BA=0.79, SD=2.22*10⁻¹⁶), and a sensitivity of 70.8% and specificity of 95%. (line 139-142)

4. Missing data was imputed. Please describe more clearly what data was imputed, and in which analysis imputed data was used. It is essential that in the test data set, over which the performance of the classification model was measured, no imputed data was used for the labels (i.e. current smokers vs non-smokers). In addition, the variables tested for association with pSIN should be free from imputation or at least it should be tested if the association holds when

imputation is not performed.

Response: Please see our response to Reviewer 1's 14th comments on the same issue. Only co-variables of the linear association between biomarkers and pSIN, clinical risk factors and pSIN, GWAS, exposome analysis and multivariable Cox model were imputed. No exposure or outcomes were imputed. No imputed data was used in the classification model. (lines 534-537)

Minor issues:

1. In the abstract, the added value of pSIN with respect to self reported smoking history, and the methylation profile should be made more clear.

Response: We have stated that the association between pSIN and diseases was performed stratified by smoking status and adjusted by other smoking histories (lines 45-48). However, as mentioned earlier we do not have methylation information on the participants to include that.

2. In the abstract, the rationale behind the genetic variant associations is completely lost. (Why would one expect a link between a lifestyle choice of smoking and genetics?). Perhaps take this out if there is no sufficient space to explain this. It is very confusing to report this without appropriate context.

Response: The purpose of GWAS is to understand the genetic architecture underlying the biological consequences of smoking pathology, as indexed by pSIN. Specifically, we sought to investigate how genetic factors contribute to the proteomic signature of smoking. We have added sentences to state more clearly the rationale of the GWAS analysis (lines 47-49).

3. Objectivity of signature:

a. Page 3, line 71/72 "... there is an urgent need to develop objective measures ..."

b. page 13 line 272 "This suggests pSIN can serve as an objective test ..."

Making claims about objectivity, should be accompanied by a discussion on the accuracy of the model. What if this signature is used to check someone's smoking history for insurance purposes? What guarantee can be provided that someone with a high pSIN score has indeed smoked? Make the limitations of the model more clear, when objectivity of the model is claimed.

Response: As with any predictive model, the accuracy of pSIN depends on the quality of the data and the model's generalizability. Although we have demonstrated high predictive accuracy (AUC = 0.95) in the UKB and CKB datasets, there are potential

limitations, particularly when applying the signature to new populations or for specific uses (e.g., insurance assessments). Although we demonstrated that a high pSIN score is strongly associated with smoking-related biological changes, it does not guarantee that someone with a high score has indeed smoked. Other factors, including genetic predisposition, environmental exposures, and other health conditions, could influence proteomic profiles. Hence its use as a definitive marker of smoking history should be approached with caution.

We have revised the text to clarify these limitations and emphasize the need for further validation and careful interpretation when considering pSIN for specific applications, such as insurance or clinical risk assessment. These limitations are now addressed in both the discussion and the conclusion to provide a more balanced view of the model's capabilities and potential uses. lines 484-491

4. “age” and “sex” are regressed out of the proteome expression data. As this is done via a linear model, and the machine learning model (gradient boosting) is non-linear, one cannot assume that there is no remaining signal left in the data to predict age or sex on the regression data. One way to test if the signal is taken out completely, is by training a classifier to predict sex or age on the regressed data set. Alternatively, one can discuss the impact on the results and signature if the regression does not remove all associations with age and sex in the proteome expression data.

Response: To address the concern regarding the potential residual signal of age and sex in the proteomic expression data after linear regression, we have performed additional analyses to rigorously evaluate the effectiveness of the regression process. Specifically, we trained gradient boosting models to assess whether any predictive signal for age or sex remained in the regressed data.

For age, we trained regression models to predict age using each protein in the regressed dataset. The results showed a mean R^2 of -0.0042 (SD = 0.0089), indicating that the models performed no better than random chance. This suggests that the linear regression successfully removed age-related signals from the proteomic data. For sex, we trained classification models to predict sex using each protein in the regressed dataset. The models achieved a mean AUC of 0.51 (SD = 0.017), which is equivalent to random guessing. This confirms that sex-related signals were effectively removed by the regression process.

These results demonstrate that the linear regression procedure successfully eliminated the influence of age and sex from the proteomic expression data. Consequently, we can confidently conclude that any associations identified in our downstream analyses are unlikely to be confounded by these variables. We have added these findings to the revised manuscript to provide further transparency and support for our approach. lines 565-575

5. Please explain how the model probability score for a given class that comes out of the classification model is transformed to a pSIN score with a range of

approximately -10.0-10.0. Make in the text of the results section clearer what a pSIN score of 0.0 indicates, or more generally how the reader can quantitatively interpret the pSIN score.

Response: The pSIN score is derived from the raw output of the LightGBM gradient boosting model, which represents the sum of the leaf values across all trees in the model. For binary classification tasks, this raw score corresponds to the log odds of the positive class (in this case, being a current smoker). The LightGBM model typically outputs raw scores (logits) in the range of approximately -10 to 10, where a score of 0 indicates a neutral prediction, corresponding to a 50% probability of being in the positive class ($\text{sigmoid}(0) = 0.5$). Scores closer to 10 indicate a high confidence prediction for the positive class ($\text{sigmoid}(10) \approx 0.99995$, or $\sim 99.995\%$ probability), while scores closer to -10 indicate a high confidence prediction for the negative class ($\text{sigmoid}(-10) \approx 4.5 \times 10^{-5}$, or $\sim 0.0045\%$ probability).

In our analysis, we set the classification threshold at a raw score of -1.29, which corresponds to the point where the false positive rate (FPR) is 0.05. This threshold was chosen to balance sensitivity and specificity in our predictions. pSIN higher than the threshold indicates a higher likelihood of being in the positive class (smoker), with larger values reflecting greater confidence, while pSIN smaller than the threshold indicate a higher likelihood of being in the negative class (non-smoker), with more negative values reflecting greater confidence. We have added this in the method section lines 656-669.

6. Very strong associations between pSIN and haematological measurements and the blood biochemical composition are found (S4 and S5). Is there any clear link between the proteins found in the plasma based pSIN signature, and these findings? Would any of these measures by itself be sufficient to classify “current smokers” and “never-smokers”.

Response: The purpose of the analysis is to investigate if pSIN is reflective of different functions represented by those biomarkers and if pSIN provides added value in addition to self-report smoking status.

The proteins included in the pSIN signature reflect a broad spectrum of biological processes, including immune regulation, cell proliferation, and inflammation, which are known to be influenced by smoking. Many of the haematological and biochemical measures (e.g., inflammatory markers such as GlycA and CRP, lipid profiles, and kidney function biomarkers) are direct or indirect indicators of the systemic impact of smoking. For example, pSIN proteins related to immune activation, such as IL12B and MMP12, are closely linked to inflammatory pathways, which are often altered in response to smoking. These inflammatory changes are reflected in biomarkers such as GlycA and CRP, which we observed to be strongly associated with pSIN levels. Additionally, proteins involved in vascular remodeling and metabolic processes, such as ACVRL1 and APOA, are associated with biochemical markers like triglycerides and HDL cholesterol, which are known to be affected by smoking.

While the haematological and biochemical measures we identified show significant associations with pSIN, none of these measures in isolation would be sufficient to reliably classify "current smokers" and "never smokers." This is because smoking has broad and complex effects on multiple biological systems, which are reflected in the proteomic signature. In contrast, the individual biomarkers may capture only specific aspects of smoking-related changes (e.g., inflammation or lipid metabolism) and do not provide a comprehensive view of the cumulative biological impact of smoking. pSIN, with its multi-protein signature, captures this broader biological context, offering a more accurate and nuanced classification of smoking status. In addition, we tested the prediction power of haematological measurements against smoking status in UKB using a gradient boosting model and in the testing dataset the AUC is 0.741 (Std=0.008).

7. P6 line 178-183. It is unclear what the rationale is for performing these genetic correlations. Are you looking for risk factors to start smoking? Or are you interested to see which genetic variants can worsen the effects of smoking, or perhaps cause a smoking-like signature in non-smokers? Make the aim of the analysis more clear, and justify the choice of datasets to ask this question.

Response: We apologize for the lack of clarity. The fundamental reason for including genetic associations in this analysis is to understand the genetic architecture underlying the biological consequences of smoking pathology, as indexed by pSIN. Specifically, we sought to investigate how genetic factors contribute to the proteomic signature of smoking. The goal is to identify potential genetic variants that influence the biological effects of smoking or exacerbate the health consequences of smoking, as well as whether certain genetic variants might contribute to a "smoking-like" biological signature in non-smokers.

We utilized the GWAS Catalog, which includes publicly available datasets on smoking behaviour and related health outcomes, to examine how genetic variants associated with pSIN either directly overlap or correlate with those linked to other diseases or traits influenced by smoking. By using these datasets, we aimed to capture both the genetic predispositions for smoking behaviour and the genetic variants that influence the biological consequences of smoking, thus building a comprehensive understanding of the genetic architecture underlying pSIN and its implications for smoking-related diseases. We have added the rationale for performing GWAS of pSIN to the revised version of the manuscript, lines 184-187

8. It may be helpful to the reader to explain better how the "smoking history" has been collected. Is this self-reported? P8 line 226: explain more clearly why it makes sense to look at variance explained from "smoking history" in a profile that was learned to classify current smokers vs never-smokers.

Response: Smoking history was primarily collected through self-reported touch-screen questionnaires in the UK Biobank and the CKB cohort. In these datasets, participants were asked to report their smoking status, the number of cigarettes smoked per day, the age at which they started smoking, and the age at which they stopped (if applicable) through a questionnaire as indicated in the method section line 533, 594

The reason for assessing the variance explained by "smoking history" in the pSIN profile is to understand the extent to which smoking-related biological effects are captured by the pSIN model, which was specifically trained to distinguish between current smokers and never smokers. Although the model was trained to classify current smokers vs. never smokers, examining variance explained by smoking history allows us to assess how much of the proteomic signature is attributable to smoking behaviour versus other variables that might influence the biological markers, such as genetics or other environmental factors.

We have clarified these points in manuscript lines 362-365 to provide a better explanation for the rationale behind this analysis and the relevance of self-reported smoking history in relation to pSIN.

9. P22 “Feature interpretation and selection” it should be clearly described over which data the feature selection and hyperparameter tuning was performed. Figure 1 suggests that this is done over the train/validation set and not over the hold-out test set; as it should be. However, this should be described explicitly in the text as well.

Response: We have now explicitly stated in the manuscript that feature selection and hyperparameter tuning were performed on the training/validation set, and not the hold-out test set, as Figure 1 suggests.

To clarify, the feature selection process using methods like the Boruta algorithm and hyperparameter tuning was carried out on the training set during the model development phase. This ensures that the model is optimized before testing on the hold-out set, which was reserved exclusively for model evaluation to avoid data leakage and ensure an unbiased performance assessment.

We have updated the text in method section lines 634-647 to explicitly describe this approach and provide more clarity regarding the data used for feature selection and hyperparameter tuning.

Reviewer #4 (Remarks on code availability):

The published code is not sufficient to redo the analysis. The readme file is not

sufficient. It is not clear which input files are required to run the script. The code is poorly annotated / commented. Internal code review is required, before resubmission.

Response: With new analyses to address the issues raised by reviewers, we will provide an updated and more readable version of the code upon acceptance of the paper.

RESPONSES TO REVIEWER COMMENTS

Reviewer #1 (Remarks to the Author):

The authors have addressed all comments nicely. I only have one outstanding comment relating to my previous comment #8. Authors have performed additional sensitivity analyses adjusting associations between biomarkers or risk factor and pSIN by smoking status, and conclude : “while pSIN captures smoking-related biological signatures, it also provides additional information beyond conventional smoking status, particularly for certain biomarkers and health outcomes”. It would be important to understand to what extent pSIN is capturing other biological factors if there were to be considered for any downstream translational application. Therefore, I strongly suggest authors perform a more in-depth analysis looking at the proportion of pSIN variance explained by a comprehensive set of biological (prevalent disease status, clinical biomarkers, anthropometric, lifestyle (including smoking status), dietary and demographic factors), genetic (e.g. PRS for smoking, COPD, lung cancer, etc.) and technical variables (potentially affecting proteome measurements, e.g. fasting time, study centre, etc.), when all variables compete in the same model.

Response: As suggested by the reviewer, we have now performed a linear model followed by ANOVA to estimate the proportion of pSIN variance explained by each category of variables (partial R^2), stratified by smoking status (**Fig S7, lines280-294**). In the overall UKB population, smoking history accounted for the largest proportion of variance (partial $R^2 = 42.5\%$), followed by clinical biomarkers (5.3%). In contrast, prevalent disease status and polygenic risk scores explained only 0.1% and 0.02% of the variance, respectively. Technical factors—including fasting time, assessment centre, and assay batch—contributed just 0.3% of the variance, underscoring the robustness of the model. Among the current smokers, clinical biomarkers explained the largest proportion of variance in current smokers (9.9%), followed by smoking history variables other than smoking status (7.8%) and lifestyle factors (3.7%). Clinical biomarkers also remained important contributors to pSIN in previous smokers (4.1%) and never smokers (4.2%). Smoking history explained the greatest variance (6.9%) among previous smokers, whereas in never smokers, it accounted for only 0.03% (passive smoking). Finally, technical factors contributed modestly to pSIN variance in current (1.4%), previous (0.4%), and never smokers (0.3%), underscoring the robustness of our model.

Detailed methodology was updated in the method section (lines 775-787).

Figure s7 Relative contribution of each category to pSIN when competing in one model.

Linear regression followed by ANOVA was performed to study the contribution of each category to pSIN stratified by smoking status. Partial R^2 was shown, and the exact percentage was annotated if it is larger than 1%. For smoking history, smoking status and pack years were included for the whole population, pack years, smoking years, tobacco type, number of cigarettes per day were included for current smokers, pack years, smoking years, smoking cessation years, tobacco type, number of cigarettes per day were included for previous smokers, and passive smoking exposure was included for never smokers.

Reviewer #4 (Remarks to the Author):

The clarity of the manuscript has much improved due to the revisions. In particular, the value and impact of the pSIN profile are much more explicitly described, and the added value is more clearly demonstrated by the additional analyses.

One minor issue remains:

Comment Reviewer 1, comment 3:

The background of the GO enrichment analysis may still be problematic. Olink panels are not a set of randomly chosen proteins (unlike RNAseq or untargeted MS proteomics), but are a selected set of proteins based on disease profiles. ClusterProfiler assumes an untargeted omics set. It would be good, to use a method, that can take the Olink target panel as background into account, otherwise pathways may come up, that were simply selected within the Olink panels.

Response: We apologise for the lack of clarity. While we understand the concern of the reviewer, we would like to point out that we have not performed a pathway enrichment analysis

of the identified proteins in the pSIN. We have only used the algorithm to annotate the function of the individual proteins. We hope this clarifies any confusion. We have also clarified in the Methods section (lines 708-709).

Reviewer #4 (Remarks on code availability):

Some parts of the code are still quite poorly annotated, but the overall structure is now clear.

Response: More annotations added to the code to make it more readable.